# DELVING INTO SPECTRAL CLUSTERING WITH VISION-LANGUAGE REPRESENTATIONS

**Bo Peng,**[*] **Yuanwei Hu,**[*] **Bo Liu, Ling Chen, Jie Lu, Zhen Fang**[†]
Australian Artificial Intelligence Institute, University of Technology Sydney, Australia
{bo.peng-7, yuanwei.hu}@student.uts.edu.au
{bo.liu, ling.chen, jie.lu, zhen.fang}@uts.edu.au

## ABSTRACT

Spectral clustering is known as a powerful technique in unsupervised data analysis. The vast majority of approaches to spectral clustering are driven by a single modality, leaving the rich information in multi-modal representations untapped. Inspired by the recent success of vision-language pre-training, this paper enriches the landscape of spectral clustering from a single-modal to a multi-modal regime. Particularly, we propose Neural Tangent Kernel Spectral Clustering that leverages cross-modal alignment in pre-trained vision-language models. By anchoring the neural tangent kernel with positive nouns, i.e., those semantically close to the images of interest, we arrive at formulating the affinity between images as a coupling of their visual proximity and semantic overlap. We show that this formulation amplifies within-cluster connections while suppressing spurious ones across clusters, hence encouraging block-diagonal structures. In addition, we present a regularized affinity diffusion mechanism that adaptively ensembles affinity matrices induced by different prompts. Extensive experiments on **16** benchmarks—including classical, large-scale, fine-grained and domain-shifted datasets—manifest that our method consistently outperforms the state-of-the-art by a large margin. Code: ⬤

## 1 INTRODUCTION

Clustering aims to partition a set of unlabeled samples into groups such that samples within the same group are semantically similar. Among various clustering techniques, spectral clustering has demonstrated superior effectiveness thanks to its ability to capture non-linear pairwise affinities. By reformulating clustering as a graph-partitioning problem, spectral clustering represents samples as nodes and pairwise affinities as edge weights. Leveraging the spectrum of the graph Laplacian, low-dimensional embeddings are then learned to reveal cluster structures. Despite these theoretical advantages, most existing approaches remain confined to visual-only representations. As a result, they often suffer from inherent limitations when semantically distinct images are visually similar, thereby yielding sub-optimal affinity graphs and degraded clustering quality.

This paper explores a new landscape for spectral clustering by moving beyond the classical single-modality paradigm toward a multi-modal regime. While the motivation is appealing, a core challenge arises: how to effectively utilize joint vision-language features for spectral clustering? In the visual domain, existing methods typically require discriminative feature representations (Shaham et al., 2018; Yang et al., 2019; Duan et al., 2019) and a distance metric (Zhang et al., 2021b; Guo et al., 2025; Li et al., 2022a), under which within-cluster images are relatively far from between-cluster images. However, such approaches do not directly translate into the multi-modal regime, where semantic alignment between modalities plays a decisive role.

On the representation learning side, the emergence of large-scale vision–language pre-training, such as CLIP (Radford et al., 2021), provides a powerful alternative to purely visual encoders. By mapping textual and visual inputs into a unified hyperspherical embedding space, CLIP captures cross-modal correspondences to enrich the semantic structure of image representations. Building on this

---

[*]Equal Contribution
[†]Correspondence to Zhen Fang (zhen.fang@uts.edu.au)

capacity, recent works (Li et al., 2024; Cai et al., 2023) select positive nouns[1] from large-scale lexical databases in the wild[2] to serve as semantic anchors in the absence of class-name priors. Despite promising potential, spectral clustering based on such aligned multi-modal features remains underexplored due to the challenge of designing a principled framework to integrate textual semantics with visual similarities in constructing an affinity matrix.

This paper addresses this research gap from a novel perspective of Neural Tangent Kernel (NTK) (Jacot et al., 2018). Our method capitalizes on the compatibility between visual and textual features. By anchoring a proxy network with features of the filtered positive nouns and computing its NTK on pairs of image features, we formulate the affinity between two images as a multiplicative coupling of (i) their visual proximity in the CLIP feature space and (ii) their semantic overlap measured by how strongly and consistently each image aligns to the positive nouns. Our theoretical analysis reveals that this coupling enhances within-cluster affinities (high visual proximity and shared semantics) and suppresses cross-cluster links (visual proximity alone is insufficient), thus sharpening the block-diagonal structure acknowledged by spectral clustering. Moreover, we present a Regularized Affinity Diffusion (RAD) mechanism to adaptively ensemble various affinity matrices induced by different prompts. In particular, RAD allows for a robust affinity matrix to be constructed through a joint optimization of ensemble weights and the equivalent objective of the diffusion process.

Extensive experiments on **16** benchmarks empirically demonstrate the effectiveness of our proposed method. For example, our method achieves 98.3% ACC and 84.9% ACC on STL-10 and ImageNet-Dogs, respectively, outperforming the latest TAC (Li et al., 2024) by 3.8% and 9.8%. In addition, on three more challenging datasets (DTD, UCF-101, and ImageNet-1K), our method surpasses TAC (Li et al., 2024) by an average of 7.7%, 2.5%, and 6.3% w.r.t. ACC, NMI, and ARI, respectively. We also validate our method in fine-grained and domain-shifted settings, and ours significantly outperforms TAC (Li et al., 2024) by 5.1% on Pets and 5.3% on ImageNet-sketch w.r.t. ACC.

## 2 PRELIMINARY

**Notation.** We denote matrices and vectors as bold-faced uppercase and lowercase characters respectively. In the remaining of this paper, we write $\mathbf{A}[i,j]$ as the $ij$-th element of the matrix $\mathbf{A}$, $\langle \cdot, \cdot \rangle$ as the inner product, and $vec(\cdot)$ as the vectorization operator. Let $\mathbf{e}[i]$ be the $i$-th element of the vector $\mathbf{e} \in \mathbb{R}^K$ and $[K] = \{1, \ldots, K\}$, we then define $\text{softmax}_k(\mathbf{e}) = \exp(\mathbf{e}[k]) / \sum_{i \in [K]} \exp(\mathbf{e}[i])$.

**Zero-shot Classification.** Let $\mathcal{X}$ and $\mathcal{T}$ be the visual and textual input space respectively, CLIP-based models adopt a dual-stream architecture with one text encoder $f_\mathcal{T}$ and one image encoder $f_\mathcal{X}$ to map inputs of two modalities into an uni-modal hyper-spherical feature space $\mathcal{Z} = \{\mathbf{z} \in \mathbb{R}^d | \|\mathbf{z}\|_2 = 1\}$. Considering an image classification task with known classes $\{\mathbf{c}_1, \ldots, \mathbf{c}_K\}$, CLIP-based models make prediction for any input $\mathbf{x} \in \mathcal{X}$ by computing

$$\arg \max_{i=1,\ldots K} \frac{\exp\left[\tau f_\mathcal{X}(\mathbf{x})^\top f_\mathcal{T}\big(\Delta(\mathbf{c}_i)\big)\right]}{\sum_{j=1}^K \exp\left[\tau f_\mathcal{X}(\mathbf{x})^\top f_\mathcal{T}\big(\Delta(\mathbf{c}_j)\big)\right]}, \tag{1}$$

where $\tau > 0$ is a temperature, $\Delta(\mathbf{c}_i) \in \mathcal{T}$ with $\Delta(\cdot)$ as the prompt template for the input class name.

**Leveraging Unlabeled Textual Data in the Wild.** Despite remarkable effectiveness (Radford et al., 2021) and provable guarantees (Chen et al., 2023), the zero-shot paradigm in Eq. (1) relies on the prior knowledge of true class names, therefore inapplicable to the unsupervised settings. In response, advanced methods (Li et al., 2024; Cai et al., 2023) propose to select a $N$-sized set of positive nouns $\{\hat{\mathbf{c}}_1, \ldots, \hat{\mathbf{c}}_N\}$ from unlabeled "in-the-wild" textual datasets, such as WordNet (Miller, 1995).

**Spectral Clustering.** For a given image dataset $\mathcal{D}_\mathcal{X} = \{\mathbf{x}_1, \ldots \mathbf{x}_M\}$, one aims to group images into $K$ distinct clusters. Let $\mathbf{A} \in \mathbb{R}^{M \times M}$ denote an affinity matrix where the element $A[i,j]$ represents the similarity between $\mathbf{x}_i$ and $\mathbf{x}_j$, the classical method SC-Ncut (Shi & Malik, 2000) converts the clustering task as a graph cut problem given by:

$$\mathbf{Y}^\star = \arg \min_{\mathbf{Y} \in \mathbb{R}^{M \times K}} \text{tr}\left(\mathbf{Y}^\top \mathbf{L} \mathbf{Y}\right), \quad \text{s.t.} \quad \mathbf{Y}^\top \mathbf{Y} = \mathbf{I}_K, \tag{2}$$

---

[1] By definition, *positive* nouns are those semantically *relevant/similar* to *any* ID label.

[2] Generally, "in-the-wild" data are those that can be collected almost for free upon deploying machine learning models in the open world.

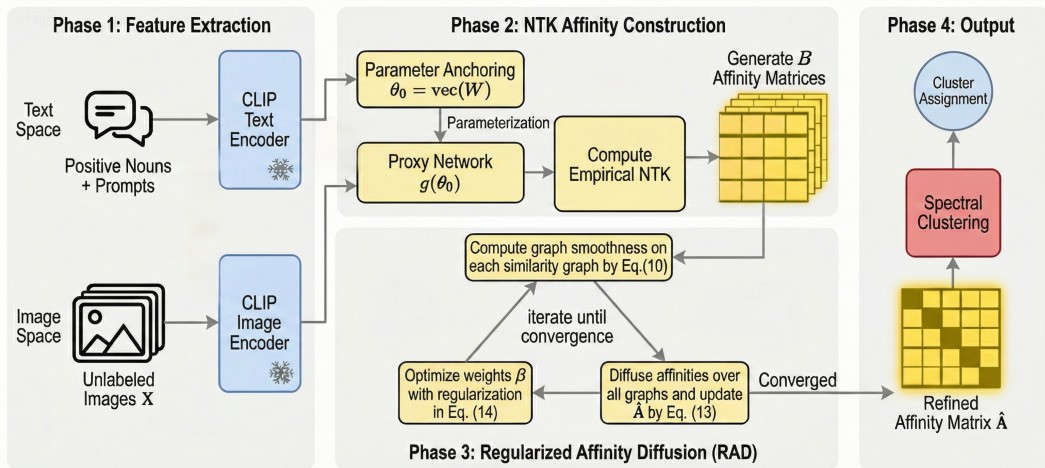

Figure 1: Overview of the proposed NTK-based spectral clustering pipeline.

where $\mathbf{I}_K$ is a $K \times K$ identity matrix, $\mathbf{L} = \mathbf{I}_M - \mathbf{D}^{-1/2}\mathbf{A}\mathbf{D}^{-1/2}$ is a normalized Laplacian matrix, $\mathbf{D}$ is a diagonal matrix of $\mathbf{D}[i,i] = \sum_j \mathbf{A}[i,j]$, and $\mathrm{tr}(\cdot)$ denotes the trace of a matrix. The optimal spectral embedding matrix $\mathbf{Y}^\star$ consists of the top-$K$ minimum eigenvectors of $\mathbf{L}$.

**Neural Tangent Kernel.** Let us define a proxy network as a function $g_{\boldsymbol{\theta}}(\cdot) : \mathcal{Z} \to \mathbb{R}$ differentiable w.r.t. parameters $\boldsymbol{\theta} \in \mathbb{R}^P$ (stretched into a single vector). In a small neighborhood region around a initialization $\boldsymbol{\theta}_0$, the function $g_{\boldsymbol{\theta}}$ can be linearly approximated by a first-order Taylor expansion:

$$g_{\boldsymbol{\theta}}(\mathbf{z}) \approx \hat{g}_{\boldsymbol{\theta}}(\mathbf{z};\boldsymbol{\theta}_0) = g_{\boldsymbol{\theta}_0}(\mathbf{z}) + \left\langle \boldsymbol{\theta} - \boldsymbol{\theta}_0, \frac{\partial g_{\boldsymbol{\theta}_0}(\mathbf{z})}{\partial \boldsymbol{\theta}_0} \right\rangle . \tag{3}$$

Eq. (3) implies that the obtained approximation $\hat{g}_{\boldsymbol{\theta}}(\mathbf{z};\boldsymbol{\theta}_0)$ can be regarded as a linearized network that maps the weights to functions residing in a reproducible kernel Hilbert space (RKHS) determined by the empirical neural tangent kernel (Jacot et al., 2018) at $\boldsymbol{\theta}_0$, i.e.,

$$\mathcal{K}_{\boldsymbol{\theta}_0}(\mathbf{z}_i, \mathbf{z}_j) = \left\langle \frac{\partial g_{\boldsymbol{\theta}_0}(\mathbf{z}_i)}{\partial \boldsymbol{\theta}_0}, \frac{\partial g_{\boldsymbol{\theta}_0}(\mathbf{z}_j)}{\partial \boldsymbol{\theta}_0} \right\rangle . \tag{4}$$

Intuitively, $\mathcal{K}_{\boldsymbol{\theta}_0}(\cdot,\cdot)$ can be interpreted as a condensed representation of gradient magnitude and gradient correlations. More concretely, the gradient magnitude governs the scale of update that each input induces on the model during optimization, while the gradient correlations dictate the alignment or stochasticity of update directions across inputs (Xu et al., 2021). Since $\mathcal{K}_{\boldsymbol{\theta}_0}(\cdot,\cdot)$ directly quantifies how two inputs interact through the dynamics of function learning, this naturally yields a higher-order notion of affinity in function space rather than their geometric closeness in the input space of $g_{\boldsymbol{\theta}}$. In this rest of this paper, we abbreviate empirical Neural Tangent Kernel as NTK for simplicity.

## 3 METHODOLOGY

**Motivation.** Eq. (2) implies that the effectiveness of spectral clustering (SC) fundamentally depends on the quality of the affinity matrix $\mathbf{A}$. To enhance the expressiveness of the affinity matrix $\mathbf{A}$, prior works (Zhong et al., 2021; Lu et al., 2024; Huang et al., 2019b) compute the affinity matrix $\mathbf{A}$ by applying kernel tricks on the latent representation space. Taking the widely used RBF kernel as an example, with pre-trained CLIP-based models, the affinity matrix can be formulated as follows:

$$\mathbf{A}_{\mathrm{RBF}}[i,j] = \begin{cases} e^{\Phi\left(f_{\mathcal{X}}(\mathbf{x}_i), f_{\mathcal{X}}(\mathbf{x}_j)\right)/\tau} & \text{if } f_{\mathcal{X}}(\mathbf{x}_i) \in \mathcal{N}_q(f_{\mathcal{X}}(\mathbf{x}_j), \Phi) \wedge f_{\mathcal{X}}(\mathbf{x}_j) \in \mathcal{N}_q(f_{\mathcal{X}}(\mathbf{x}_i), \Phi), \\ 0 & \text{otherwise.} \end{cases} \tag{5}$$

where $\Phi(\mathbf{z}_i, \mathbf{z}_j) = -\|\mathbf{z}_i - \mathbf{z}_j\|_2^2$ and $\mathcal{N}_q(\mathbf{z}, \Phi)$ represents top-$q$ nearest neighbors of $\mathbf{z}$ with regard to the metric function $\Phi(\cdot, \mathbf{z})$. While Table 1 shows that CLIP(SC), which uses the precomputed $\mathbf{A}_{\mathrm{RBF}}$ in Eq. (5) for SC, outperforms most traditional training-based competitors[3], CLIP(SC) relies

---

[3]Experimental results using other kernels are provided in appendix (c.f. Table 6)

on single-modal visual features and thus leaves the semantic knowledge embedded in text largely unexploited. In response to this, a naive strategy is to perform the RBF-based affinity measure on the image-text-concentrated feature space from TAC (Li et al., 2024) for SC. Unfortunately, our experimental results (c.f. Table 1 and Table 2) reveal that TAC(SC) yields performance comparable to TAC(KMeans). This observation means that merely incorporating text at the feature level does not adequately guide affinity construction, prompting the research question underlying this work:

> *How can we effectively integrate textual semantics with visual similarities in constructing the affinity matrix* $\mathbf{A}$*?*

## 3.1 NEURAL TANGENT KERNEL SPECTRAL CLUSTERING

To address this challenge, we present a novel spectral clustering framework build upon the NTK in Eq. (4). To implement our method, we begin with extracting features from CLIP-based models for the observed positive nouns $\{\hat{\mathbf{c}}_1, \ldots, \hat{\mathbf{c}}_N\}$ to have $\mathbf{W} = [\mathbf{w}_1, \ldots, \mathbf{w}_N] \in \mathbb{R}^{d \times N}$ where $\mathbf{w}_i = f_{\mathcal{T}}(\Delta(\hat{\mathbf{c}}_i))$ for each $i \in [N]$. Thanks to the cross-modal alignment of pre-trained CLIP-based models, we subsequently take $\boldsymbol{\theta}_0 = vec(\mathbf{W})$ to anchor the NTK in a semantically meaningful parameterization. This enables the gradient $\partial g_{\boldsymbol{\theta}_0}(\mathbf{z})/\partial \boldsymbol{\theta}_0$ to capture how the input $\mathbf{z}$ functionally interacts with positive nouns, hence injecting the learned semantic structure of texts from CLIP-based models into the NTK-based affinity matrix $\mathbf{A}_{\text{NTK}}$ that is given as follows:

$$\mathbf{A}_{\text{NTK}}[i, j] = \begin{cases} \mathcal{K}_{\boldsymbol{\theta}_0}(f_{\mathcal{X}}(\mathbf{x}_i), f_{\mathcal{X}}(\mathbf{x}_j)) & \text{if } f_{\mathcal{X}}(\mathbf{x}_i) \in \mathcal{N}_q(f_{\mathcal{X}}(\mathbf{x}_j), \mathcal{K}_{\boldsymbol{\theta}_0}) \wedge f_{\mathcal{X}}(\mathbf{x}_j) \in \mathcal{N}_q(f_{\mathcal{X}}(\mathbf{x}_i), \mathcal{K}_{\boldsymbol{\theta}_0}), \\ 0 & \text{otherwise.} \end{cases}$$

(6)

While the formulation of $\mathcal{K}_{\boldsymbol{\theta}_0}(\cdot, \cdot)$ is generic to the choice of $g_{\boldsymbol{\theta}_0}(\cdot)$, when applying it to spectral clustering, this paper proposes to design $g_{\boldsymbol{\theta}_0}(\cdot)$ in a log-sum-exp form, i.e.,

$$g_{\boldsymbol{\theta}_0}(f_{\mathcal{X}}(\mathbf{x}_i)) = \log \sum_{k=1}^{N} e^{\mathbf{w}_k^\top f_{\mathcal{X}}(\mathbf{x}_i)/\tau}.$$

(7)

In the following, we provide a theoretical justification of Eq. (7) by showing that it can promote a block structure in the constructed affinity matrix.

In particular, combining Eq. (7) with Eq. (4), the closed-form formulation of $\mathcal{K}_{\boldsymbol{\theta}_0}(\cdot, \cdot)$ is given by

$$\mathcal{K}_{\boldsymbol{\theta}_0}(f_{\mathcal{X}}(\mathbf{x}_i), f_{\mathcal{X}}(\mathbf{x}_j)) = \frac{1}{\tau^2} \cdot \underbrace{f_{\mathcal{X}}(\mathbf{x}_i)^\top f_{\mathcal{X}}(\mathbf{x}_j)}_{U_{ij}} \cdot \underbrace{\left(\sum_{k=1}^{N} \mathbf{s}_i[k]\mathbf{s}_j[k]\right)}_{V_{ij}},$$

(8)

where $\mathbf{s}_i[k] = \text{softmax}_k(\mathbf{W}^\top f_{\mathcal{X}}(\mathbf{x}_i)/\tau)$. Since $\tau$ is usually set to a relatively small value (e.g., $\tau = 0.04$ in this paper), each softmax probability $\mathbf{s}_i$ can be highly skewed in practice.

From Eq. (8), one can find that $\mathcal{K}_{\boldsymbol{\theta}_0}(f_{\mathcal{X}}(\mathbf{x}_i), f_{\mathcal{X}}(\mathbf{x}_j))$ is multiplicative coupling of visual proximity $U_{ij}$ and semantic overlap $V_{ij}$. For images within the same cluster, both terms tend to be simultaneously large—their embeddings are close in the CLIP space and their softmax probabilities concentrate on positive nouns—so their affinities are strongly amplified, filling the diagonal blocks with high values. For images across different clusters, even if $U_{ij}$ is moderate due to visual similarity, their softmax probabilities would concentrate on a different subset of positive nouns, yielding a sufficiently small $V_{ij}$ and therefore suppressing cross-cluster affinities. This multiplicative mechanism enforces high intra-cluster and low inter-cluster similarity, thereby sharpening the block-diagonal pattern of the NTK-induced affinity matrix $\mathbf{A}_{\text{NTK}}$.

## 3.2 AFFINITY ENSEMBLING VIA REGULARIZED AFFINITY DIFFUSION

In this section, we extend our method to the same multi-prompt scenario as TAC (Li et al., 2024)[4]. Formally, given a $B$-sized pool of prompt templates $\{\Delta^{(1)}, \ldots, \Delta^{(B)}\}$, we can construct a prompt-specific affinity matrix $\mathbf{A}_{\text{NTK}}^{(b)}$ for each $\Delta^{(b)}$ with $b \in [B]$. In view of this, our goal is to learn a

---

[4]Although the TAC paper states that a single prompt template is used, its official implementation in fact employs multiple prompt templates indeed.

new affinity matrix $\hat{\mathbf{A}}$ which 1) captures the geometry of the underlying manifold encoded in each affinity matrix, and 2) leverages the complementarity among multiple prompt templates.

To this end, the most straightforward strategy is to simply average the $B$ affinity matrices $\mathbf{A}_{\text{NTK}}^{(1)}, \ldots, \mathbf{A}_{\text{NTK}}^{(B)}$, i.e., $\hat{\mathbf{A}} = \frac{1}{B} \sum_{b=1}^{B} \mathbf{A}_{\text{NTK}}^{(b)}$. Despite simplicity, this naive practice ignores the correlations among different affinities as it assigns a uniform weight to each affinity matrix. Fortunately, Bai et al. (2016) show that weight learning can be exerted on affinity matrices to assist neighborhood structure mining. This motivates us to formulate the weight learning and the affinity learning as a unified optimization problem given by

$$\min_{\boldsymbol{\beta}, \hat{\mathbf{A}}} \sum_{b=1}^{B} \boldsymbol{\beta}[b] \cdot \ell(\hat{\mathbf{A}}, \mathbf{A}_{\text{NTK}}^{(b)}) + \mu \|\hat{\mathbf{A}} - \mathbf{E}\|_F^2 + \frac{\lambda}{2} \|\boldsymbol{\beta}\|_2^2 \quad \text{s.t.} \quad 0 \leq \boldsymbol{\beta}[b] \leq 1 \text{ and } \sum_{b=1}^{B} \boldsymbol{\beta}[b] = 1, \quad (9)$$

where $\mathbf{E}$ is a positive semi-definite matrix[5] to avoid $\hat{\mathbf{A}}$ from being extremely smoothed[6], $\boldsymbol{\beta}[b]$ denotes the contribution of $\mathbf{A}_{\text{NTK}}^{(b)}$ to the whole affinity diffusion process, and

$$\ell(\hat{\mathbf{A}}, \mathbf{A}_{\text{NTK}}^{(b)}) = \frac{1}{2} \sum_{i,j,k,l=1}^{M} a_{ij}^{(b)} a_{kl}^{(b)} \left( \frac{\hat{\mathbf{A}}[k, i]}{\sqrt{d_i^{(b)} d_k^{(b)}}} - \frac{\hat{\mathbf{A}}[l, j]}{\sqrt{d_j^{(b)} d_l^{(b)}}} \right) \quad (10)$$

is the objective value of the affinity diffusion process (Bai et al., 2018; 2017) with $a_{ij}^{(b)} = \mathbf{A}_{\text{NTK}}^{(b)}[i, j]$ and $d_i^{(b)} = \sum_{j=1}^{M} \mathbf{A}_{\text{NTK}}^{(b)}[i, j]$.

Note that the optimization problem in Eq. (9) simultaneously depends on $\boldsymbol{\beta}$ and $\hat{\mathbf{A}}$, therefore making it inherently complex and impractical to find a direct solution. To resolve this, we propose a numerical method that decomposes the optimization problem into two sub-problems: 1) *Optimize $\hat{\mathbf{A}}$ with Fixed $\boldsymbol{\beta}$* and 2) *Optimize $\boldsymbol{\beta}$ with Fixed $\hat{\mathbf{A}}$*. This allows for a systematic iterative approximation of the optimal result through the fixed-point scheme.

**Optimize $\hat{\mathbf{A}}$ with Fixed $\boldsymbol{\beta}$.** In this scenario, as we show in Appendix B, the optimization problem in Eq. (9) can be transformed as follows:

$$\min_{\hat{\mathbf{A}}} \sum_{b=1}^{B} \boldsymbol{\beta}[b] \cdot vec(\hat{\mathbf{A}})^{\top} (\mathbf{I}_{M^2} - \mathbb{S}^{(b)}) vec(\hat{\mathbf{A}}) + \mu \|vec(\hat{\mathbf{A}}) - vec(\mathbf{E})\|_2^2, \quad (11)$$

where $\mathbb{S}^{(b)} = \mathbf{S}^{(b)} \otimes \mathbf{S}^{(b)} \in \mathbb{R}^{M^2 \times M^2}$ is the Kronecker product of $\mathbf{S}^{(b)}$ and itself with $\mathbf{S}^{(b)}$ as the row-normalized $\mathbf{A}_{\text{NTK}}^{(b)}$. As we show in Appendix C, the closed-form of the solution to Eq. (11) can be given as follows:

$$\hat{\mathbf{A}}^* = \frac{\mu}{\mu + 1} \cdot vec^{-1} \left( \left( \mathbf{I}_{M^2} - \sum_{b=1}^{B} \frac{\boldsymbol{\beta}[b]}{\mu + 1} \mathbb{S}^{(b)} \right)^{-1} vec(\mathbf{E}) \right), \quad (12)$$

where $vec^{-1}(\cdot)$ is the inverse operator of $vec(\cdot)$.

Since it is computationally infeasible to directly solve the inverse of a $M^2 \times M^2$ matrix, we alternatively resort to an efficient iteration-based solver, i.e.,

$$\hat{\mathbf{A}} \leftarrow \sum_{b=1}^{B} \frac{\boldsymbol{\beta}[b]}{\mu + 1} \mathbf{S}^{(b)} \hat{\mathbf{A}} \mathbf{S}^{(b)\top} + \frac{\mu}{\mu + 1} \mathbf{E}. \quad (13)$$

Appendix D proves the convergence of the iterative process in Eq. (13) to the solution in Eq. (12).

**Optimize $\boldsymbol{\beta}$ with Fixed $\hat{\mathbf{A}}$.** In this case, the optimization problem in Eq. (9) can be simplified as

$$\min_{\boldsymbol{\beta}} \sum_{b=1}^{B} \boldsymbol{\beta}[b] \cdot \ell(\hat{\mathbf{A}}, \mathbf{A}_{\text{NTK}}^{(b)}) + \frac{\lambda}{2} \|\boldsymbol{\beta}\|_2^2, \quad s.t. \quad 0 \leq \boldsymbol{\beta}[b] \leq 1 \text{ and } \sum_{b=1}^{B} \boldsymbol{\beta}[b] = 1. \quad (14)$$

---

[5]In practice, we find that simply setting $\mathbf{E} = \mathbf{I}_M$ helps.
[6]In this case, all vectors in $\hat{\mathbf{A}}$ are nearly identical.

Table 1: Clustering performance (%) on widely-used datasets. The best results are shown in **bold**.

| Dataset | STL-10 | | | CIFAR-10 | | | CIFAR-20 | | | ImageNet-10 | | | ImageNet-Dogs | | |
|---|---|---|---|---|---|---|---|---|---|---|---|---|---|---|---|
| Metrics | NMI | ACC | ARI | NMI | ACC | ARI | NMI | ACC | ARI | NMI | ACC | ARI | NMI | ACC | ARI |
| CLIP (zero-shot) | 93.9 | 97.1 | 93.7 | 80.7 | 90.0 | 79.3 | 55.3 | 58.3 | 39.8 | 95.8 | 97.6 | 94.9 | 73.5 | 72.8 | 58.2 |
| DAC | 36.6 | 47.0 | 25.7 | 39.6 | 52.2 | 30.6 | 18.5 | 23.8 | 8.8 | 39.4 | 52.7 | 30.2 | 21.9 | 27.5 | 11.1 |
| DCCM | 37.6 | 48.2 | 26.2 | 49.6 | 62.3 | 40.8 | 28.5 | 32.7 | 17.3 | 60.8 | 71.0 | 55.5 | 32.1 | 38.3 | 18.2 |
| IIC | 49.6 | 59.6 | 39.7 | 51.3 | 61.7 | 41.1 | 22.5 | 25.7 | 11.7 | – | – | – | – | – | – |
| PICA | 61.1 | 71.3 | 53.1 | 59.1 | 69.6 | 51.2 | 31.0 | 33.7 | 17.1 | 80.2 | 87.0 | 76.1 | 35.2 | 35.3 | 20.1 |
| CC | 76.4 | 85.0 | 72.6 | 70.5 | 79.0 | 63.7 | 43.1 | 42.9 | 26.6 | 85.9 | 89.3 | 82.2 | 44.5 | 42.9 | 27.4 |
| IDFD | 64.3 | 75.6 | 57.5 | 71.1 | 81.5 | 66.3 | 42.6 | 42.5 | 26.4 | 89.8 | 95.4 | 90.1 | 54.6 | 59.1 | 41.3 |
| SCAN | 69.8 | 80.9 | 64.6 | 79.7 | 88.3 | 77.2 | 48.6 | 50.7 | 33.3 | – | – | – | 61.2 | 59.3 | 45.7 |
| MiCE | 63.5 | 75.2 | 57.5 | 73.7 | 83.5 | 69.8 | 43.6 | 44.0 | 28.0 | – | – | – | 42.3 | 43.9 | 28.6 |
| GCC | 68.4 | 78.8 | 63.1 | 76.4 | 85.6 | 72.8 | 47.2 | 47.2 | 30.5 | 84.2 | 90.1 | 82.2 | 49.0 | 52.6 | 36.2 |
| NNM | 66.3 | 76.8 | 59.6 | 73.7 | 83.7 | 69.4 | 48.0 | 45.9 | 30.2 | – | – | – | 60.4 | 58.6 | 44.9 |
| TCC | 73.2 | 81.4 | 68.9 | 79.0 | 90.6 | 73.3 | 47.9 | 49.1 | 31.2 | 84.8 | 89.7 | 82.5 | 55.4 | 59.5 | 41.7 |
| SPICE | 81.7 | 90.8 | 81.2 | 73.4 | 83.8 | 70.5 | 44.8 | 46.8 | 29.4 | 82.8 | 92.1 | 83.6 | 57.2 | 64.6 | 47.9 |
| SeCu | 70.7 | 81.4 | 65.7 | 79.9 | 88.5 | 78.2 | 51.6 | 51.6 | 36.0 | – | – | – | – | – | – |
| DivClust | – | – | – | 71.0 | 81.5 | 67.5 | 44.0 | 43.7 | 28.3 | 85.0 | 90.0 | 81.9 | 51.6 | 52.9 | 37.6 |
| RPSC | 83.8 | 92.0 | 83.4 | 75.4 | 85.7 | 73.1 | 47.6 | 51.8 | 34.1 | 83.0 | 92.7 | 85.8 | 55.2 | 64.0 | 46.5 |
| TCL | 79.9 | 86.8 | 75.7 | 81.9 | 88.7 | 78.0 | 52.9 | 53.1 | 35.7 | 87.5 | 89.5 | 83.7 | 62.3 | 64.4 | 51.6 |
| ProPos | 75.8 | 86.7 | 73.7 | 85.1 | 91.6 | 83.5 | 58.2 | 57.8 | 42.3 | 89.6 | 95.6 | 90.6 | 73.7 | 77.5 | 67.5 |
| CLIP (KMeans) | 90.1 | 93.8 | 92.4 | 70.3 | 74.2 | 61.6 | 49.9 | 45.5 | 28.3 | 96.9 | 98.2 | 96.1 | 39.8 | 38.1 | 20.1 |
| CLIP (SC) | 88.1 | 87.8 | 82.4 | 65.4 | 69.2 | 57.1 | 45.2 | 42.7 | 30.1 | 96.4 | 96.8 | 95.6 | 72.8 | 70.6 | 56.9 |
| SIC | 95.3 | 98.1 | 95.9 | **84.7** | **92.6** | **84.4** | 59.3 | 58.3 | **43.9** | 97.0 | 98.2 | 96.1 | 69.0 | 69.7 | 55.8 |
| TAC (KMeans) | 92.3 | 94.5 | 89.5 | 80.8 | 90.1 | 79.8 | 60.7 | 55.8 | 42.7 | 97.5 | 98.6 | 97.0 | 75.1 | 75.1 | 63.6 |
| TAC (SC) | 92.6 | 94.3 | 94.2 | 81.2 | 90.3 | 80.1 | 56.9 | 54.5 | 30.1 | 97.0 | 98.3 | 96.8 | 75.3 | 75.8 | 64.4 |
| Gradnorm | 95.6 | **98.3** | 96.2 | 82.6 | 91.1 | 81.5 | 61.3 | **60.6** | 43.6 | **98.7** | **99.4** | **98.7** | 81.0 | 81.2 | 70.9 |
| Ours | **95.8** | **98.3** | **96.3** | 83.3 | 92.0 | 83.0 | **63.3** | 59.6 | 43.5 | 97.8 | 99.2 | 98.4 | **82.4** | **84.9** | **71.4** |

Since the optimization objective in Eq. (14) takes the form of a Lasso optimization problem, the solution to Eq. (14) can be efficiently obtained with the coordinate descent method (Wu & Lange, 2008; Wright, 2015). To keep the main text concise, we provide a detailed illustration in Appendix E.

The optimization procedure is guaranteed to converge since we obtain the optimal solution to each subproblem. By solving two subproblems alternatively, the objective value of Eq. (9) keeps decreasing monotonically, which is consistent with the visualization in Fig. 3. For clarity, the optimization procedure is summarized in appendix (c.f. Algorithm 1). Finally, we compute clustering assignment by performing spectral clustering on the ensembled affinity matrix $\hat{\mathbf{A}}$ produced by Algorithm 1.

## 4 EXPERIMENTS

**Evaluation Metric.** We evaluate clustering performance with three widely used metrics, including Accuracy (ACC), Normalized Mutual Information (NMI) and Adjusted Rand Index (ARI).

**Baselines.** We compare our method with DAC (Chang et al., 2017), DCCM (Wu et al., 2019), IIC (Ji et al., 2019), PICA (Huang et al., 2020), CC (Li et al., 2021b), IDFD (Tao et al., 2020), SCAN (Van et al., 2020), MiCE (Tsai et al., 2020), GCC (Zhong et al., 2021), NNM (Dang et al., 2021), TCC (Shen et al., 2021), SPICE (Niu et al., 2022), SeCu (Qian, 2023), DivClust (Metaxas et al., 2023), RPSC (Liu et al., 2024), TCL (Li et al., 2022b), ProPos (Huang et al., 2022), SIC (Cai et al., 2023), TAC (Li et al., 2024), GradNorm (Peng et al., 2025c).

**Implementation.** Following prior works (Li et al., 2024; Peng et al., 2025b), we adopt the pre-trained CLIP model with ViT-B/32 (Dosovitskiy et al., 2020) and Transformer (Vaswani et al., 2017) as default image and text backbones, respectively. We use the positive nouns filtered by TAC (Li et al., 2024) from WordNet (Miller, 1995). Following Li et al. (2024), we filter positive nouns based on the train split of each dataset, followed by evaluating the clustering performance on the test split of each dataset. We fix $\tau = 0.04$, $q = 30$, $\mu = 0.1$ and $\lambda = 10$ for all datasets.

**Prompt Templates.** In consistent with TAC (Li et al., 2024), we use the following $B = 7$ templates to construct prompts for the filtered positive nouns: *itap of a {}, a bad photo of the {}, a origami {}, a photo of the large {}, a {} in a video game, art of the {}, a photo of the small {}.*

Table 2: Clustering performance (%) on challenging datasets. The best results are shown in **bold**.

| Dataset | DTD | | | UCF-101 | | | ImageNet-1K | | | Average | | |
|---|---|---|---|---|---|---|---|---|---|---|---|---|
| Metrics | NMI | ACC | ARI | NMI | ACC | ARI | NMI | ACC | ARI | NMI | ACC | ARI |
| CLIP (zero-shot) | 56.5 | 43.1 | 26.9 | 79.9 | 63.4 | 50.2 | 81.0 | 63.6 | 45.4 | 72.5 | 56.7 | 40.8 |
| CLIP (KMeans) | 57.3 | 42.6 | 27.4 | 79.5 | 58.2 | 47.6 | 72.3 | 38.9 | 27.1 | 69.7 | 45.6 | 34.0 |
| CLIP (SC) | 57.9 | 46.7 | 31.8 | 81.7 | 63.3 | 55.1 | 73.9 | 40.0 | 29.8 | 71.2 | 50.0 | 38.9 |
| SCAN | 59.4 | 46.4 | 31.7 | 79.7 | 61.1 | 53.1 | 74.7 | 44.7 | 32.4 | 71.3 | 50.7 | 39.1 |
| SIC | 59.6 | 45.9 | 30.5 | 81.0 | 61.9 | 53.6 | 77.2 | 47.0 | 34.3 | 72.6 | 51.6 | 39.5 |
| TAC (KMeans) | 60.1 | 45.9 | 29.0 | 81.6 | 61.3 | 52.4 | 77.8 | 48.9 | 36.4 | 73.2 | 52.0 | 39.3 |
| TAC (SC) | 58.6 | 44.0 | 27.1 | 79.6 | 60.0 | 50.1 | 78.0 | 49.1 | 36.2 | 72.1 | 51.0 | 37.8 |
| Gradnorm | **63.1** | 50.9 | **34.2** | 82.5 | 62.7 | 53.2 | **79.2** | 52.6 | 39.1 | **74.9** | 55.4 | 41.7 |
| Ours | 61.7 | **52.0** | 33.6 | **83.0** | **67.9** | **59.4** | **79.2** | **56.3** | **39.4** | 74.6 | **58.7** | **44.1** |

## 4.1 MAIN RESULTS

**Datasets.** We evaluate the effectiveness of our method by conducting experiments over 1) five widely-used datasets: STL-10 (Coates et al., 2011), CIFAR-10 (Krizhevsky & Hinton, 2009), CIFAR-20 (Krizhevsky & Hinton, 2009), ImageNet-10 (Chang et al., 2017) and ImageNet-Dogs (Chang et al., 2017); 2) three more complex and challenging datasets: DTD (Cimpoi et al., 2014), UCF101 (Soomro et al., 2012) and ImageNet-1k (Deng et al., 2009).

**Evaluation on Classical Datasets.** Different with early baselines adopting ResNet-34 or ResNet-18 as the backbone, this paper mainly focuses on comparisons with zero-shot CLIP and CLIP-based methods. As shown in Table 1, our methods consistently outperforms TAC (Li et al., 2024) and zero-shot CLIP (Radford et al., 2021) on five classical datasets. Notably, our method achieves 7.8% and 9.8% improvement in ARI and ACC on ImageNet-Dogs, respectively. On CIFAR-10 dataset, SIC (Cai et al., 2023) is marginally superior to our method, as it entails more trainable parameters and a more sophisticated training strategy.

**Evaluation on Challenging Datasets.** Since the rapid development of pre-trained models has made clustering on relatively simple datasets such as STL-10 and CIFAR-10 no longer challenging, we evaluate our proposed method on three more challenging datasets: DTD (Cimpoi et al., 2014), UCF101 (Soomro et al., 2012) and ImageNet-1k (Deng et al., 2009). Our proposed method notably achieves the state-of-the-art perfomance which is provided in Table 2. To be specific, the proposed method outperforms TAC over 7.0% in ARI and 6.9% in ACC on UCF101. The results prove the effectiveness of our proposed strategy in studying spectral clustering from the perspective of NTK.

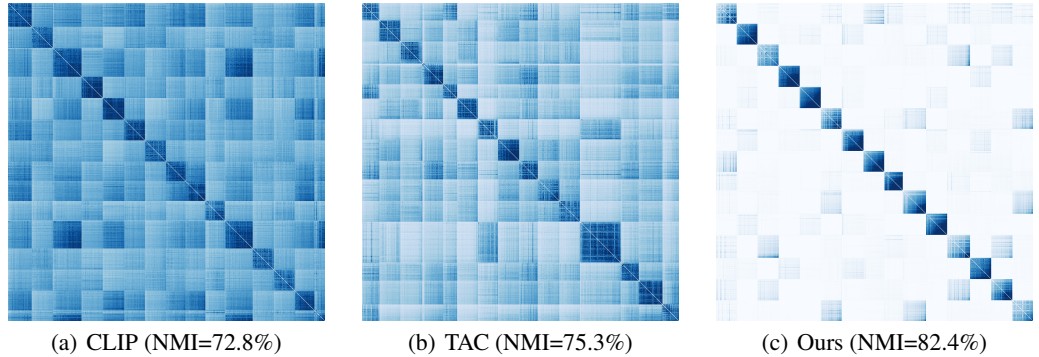

| (a) CLIP (NMI=72.8%) | (b) TAC (NMI=75.3%) | (c) Ours (NMI=82.4%) |

Figure 2: Visualization of affinity matrices on ImageNet-Dogs.

## 4.2 VISUALIZATION

**Affinity Matrix.** We examine how different choices of affinity measure impact the resulting affinity matrix in Figure 2, where rows and columns of each affinity matrix are permuted according to the ground-truth labels of images. In contrast to affinity matrices that are computed by applying the RBF kernel on the pre-trained CLIP features (Fig. 2(a)) and TAC's image-text-concentrated

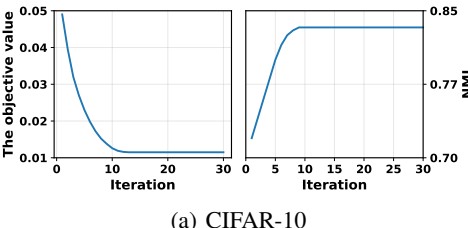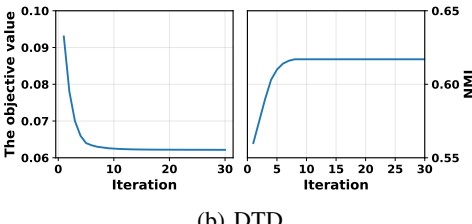

|(a) CIFAR-10|(b) DTD|
|---|---|

Figure 3: The objective value of Eq. (9) and the clustering performance (measured by NMI) at each optimization iteration on CIFAR-10 and DTD, respectively.

features (Fig. 2(b)) respectively, the affinity matrix in Fig. 2(c), which is computed by applying our proposed NTK on the pre-trained CLIP features, exhibits the sharpest block-diagonal structure: within-block entries are dense and homogeneous while off-block values are suppressed toward zero. Visualizations on other datasets can be found in appendix (c.f. Fig. 8 and Fig. 9).

**Convergence Speed.** In Fig. 3, we plot the objective value of Eq. (9) and the clustering performance at each iteration of the diffusion process. As can be clearly seen from Fig. 3, when affinities are propagated iteratively, the objective value keeps decreasing and the clustering performance keeps increasing until reaching the equilibrium. Moreover, Fig. 3 shows that the objective value of Eq. (9) converges within a small number of iterations, which implies the efficiency of our proposed method.

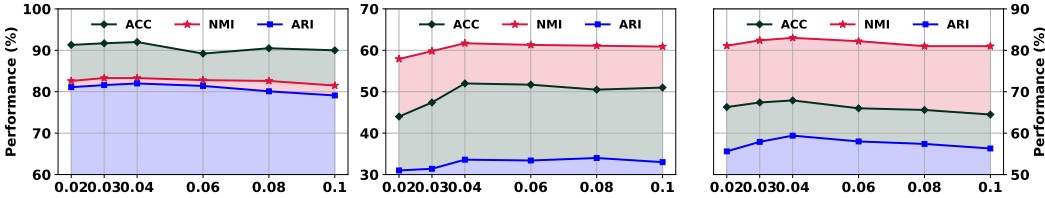

Figure 4: Ablation analysis of clustering performance by varying the value of $\tau$ on CIFAR-10 (left), DTD (middle) and UCF101 (right), respectively.

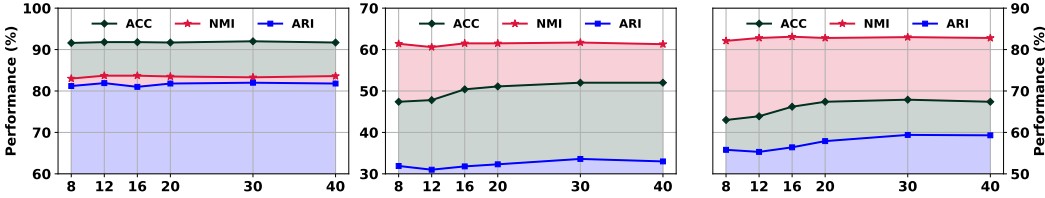

Figure 5: Ablation analysis of clustering performance by varying the value of $q$ on CIFAR-10 (left), DTD (middle) and UCF101 (right), respectively.

### 4.3 ABLATION STUDY

**Ablation on Hyper-parameters.** We evaluate the hyper-parameters most essential to our algorithm design, including the temperature $\tau$ in Eq. (6), the number of nearest neighbors $q$ in Eq. (8), and the weighting parameters $\mu$ and $\lambda$ in Eq. (9). Fig. 4 and Fig. 5 show that having a large or small value of $\tau$ and $q$ does not necessarily improve the performance. As can be found in Fig. 6 and Fig. 7, our method is stable across a wide range of the weighting parameters $\mu$ and $\lambda$.

**Ablation on Visual Encoder.** We evaluate it with two different visual encoders, ViT-B/16 and ViT-L/14, and report the clustering results in Table 3. It can be seen that the clustering performance can be enhanced by more powerful visual encoders. While our proposed method consistently outperforms TAC across both backbones, indicating better generalization.

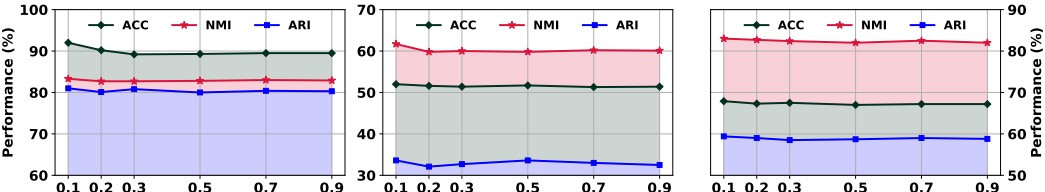

Figure 6: Ablation analysis of clustering performance by varying the value of $\mu$ on CIFAR-10 (left), DTD (middle) and UCF101 (right), respectively.

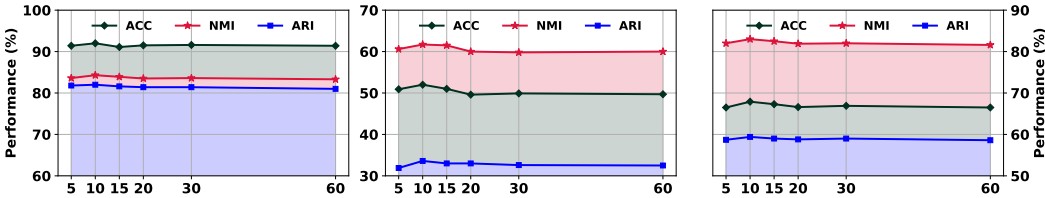

Figure 7: Ablation analysis of clustering performance by varying the value of $\lambda$ on CIFAR-10 (left), DTD (middle) and UCF101 (right), respectively.

## 4.4 DOMAIN-GENERALIZABLE IMAGE CLUSTERING.

To assess the transferability of our method, we perform clustering on several versions of ImageNet-1K variants, including ImageNet-C (Hendrycks & Dietterich, 2019), ImageNet-V2 (Kornblith et al., 2019) and ImageNet-S (Wang et al., 2019). As observed in Table 4, both TAC and our method suffer performance drops under these shifts, underscoring the challenge of clustering in such settings. Even so, our method consistently surpasses TAC across the in-distribution variants, highlighting its stronger robustness to domain shifts.

## 4.5 FINE-GRAINED IMAGE CLUSTERING.

We validate our method under the fine-grained scenario by conducting experiments on five fine-grained datasets, including Aircraft (Maji et al., 2013), Food (Bossard et al., 2014), Flowers (Nilsback & Zisserman, 2008), Pets (Parkhi et al., 2012) and Cars (Krause et al., 2013). As shown in Table 5, our proposed method consistently outperforms the state-of-the-art, which highlights the superiority of text-informed affinities for image clustering.

## 5 RELATED WORK

**Deep Spectral Clustering.** Deep spectral clustering integrates traditional spectral clustering with deep neural networks for effective clustering. Tian et al. (2014) observe the similarity between the optimization objectives of autoencoders and spectral clustering. They thus propose replacing spectral embedding with a deep autoencoder that is trained to reconstruct the pre-defined affinity matrix. However, obtaining such an affinity matrix can be challenging for complex data, and the matrix size can become prohibitively large for large datasets. SpectralNet (Shaham et al., 2018) proposes to directly embed raw data into the eigenspace of a given affinity matrix, which has inspired numerous extensions. Yang et al. (2019) enhance the robustness of SpectralNet's embeddings using a dual autoencoder. Huang et al. (2019b;a), respectively, extend SpectralNet to handle multi-view data. Another line of works (Duan et al., 2019; Affeldt et al., 2020; Golikov et al., 2022) highlight the benefits of joint optimization and propose joint spectral embedding learning and clustering.

**Image Clustering with Vision-Language Representations.** The seminar work called SIC (Cai et al., 2023) uses textual semantics to enhance image pseudo-labeling, followed by performing image clustering with consistency learning in both image space and semantic space. Note that, SIC essentially pulls image embeddings closer to embeddings in semantic space, while ignoring the im-

Table 3: Clustering performance (%) on five widely-used datasets. The best results are in **bold**.

| Backbone | Dataset | STL-10 | | | CIFAR-10 | | | CIFAR-20 | | | ImageNet-Dogs | | | DTD | | |
|---|---|---|---|---|---|---|---|---|---|---|---|---|---|---|---|---|
| | Metrics | NMI | ACC | ARI | NMI | ACC | ARI | NMI | ACC | ARI | NMI | ACC | ARI | NMI | ACC | ARI |
| ViT-B/16 | TAC (KMeans) | 95.1 | 96.1 | 93.6 | 82.9 | 91.2 | 80.5 | 62.6 | 60.4 | 45.7 | 82.3 | 81.9 | 72.9 | 62.6 | 50.4 | 33.6 |
| | TAC (SC) | 92.9 | 96.3 | 92.3 | 83.1 | 91.3 | 82.0 | 63.0 | 60.0 | 45.1 | 82.1 | 81.5 | 73.0 | 63.1 | 51.7 | 34.6 |
| | Ours | **97.2** | **99.0** | **97.4** | **86.0** | **93.1** | **85.4** | **65.7** | **64.4** | **49.1** | **84.9** | **86.7** | **75.0** | **66.3** | **55.8** | **37.2** |
| ViT-L/14 | TAC (KMeans) | 95.4 | 96.7 | 94.2 | 89.1 | 93.9 | 86.7 | 64.8 | 62.9 | 47.6 | 84.3 | 84.0 | 75.4 | 64.7 | 52.9 | 35.1 |
| | TAC (SC) | 93.8 | 96.7 | 93.8 | 89.4 | 94.1 | 88.0 | 65.0 | 63.0 | 47.5 | 84.0 | 84.0 | 75.0 | 65.1 | 53.5 | 35.9 |
| | Ours | **97.7** | **99.5** | **98.1** | **92.1** | **96.6** | **90.7** | **66.9** | **66.1** | **50.8** | **86.2** | **88.3** | **79.0** | **68.1** | **58.0** | **38.9** |

Table 4: Clustering performance (%) robustness to domain shift. The best results are in **bold**.

| Dataset | ImageNet-C | | | ImageNet-V2 | | | ImageNet-S | | | Average | | |
|---|---|---|---|---|---|---|---|---|---|---|---|---|
| Metrics | NMI | ACC | ARI | NMI | ACC | ARI | NMI | ACC | ARI | NMI | ACC | ARI |
| TAC (KMeans) | 71.4 | 39.2 | 25.6 | 71.7 | 38.5 | 23.0 | 70.7 | 34.8 | 22.1 | 71.3 | 37.5 | 23.6 |
| TAC (SC) | 70.9 | 39.0 | 25.5 | 72.0 | 39.0 | 23.6 | 70.1 | 33.7 | 21.6 | 71.0 | 37.2 | 23.6 |
| Ours | **75.0** | **44.0** | **28.1** | **74.5** | **42.7** | **27.2** | **72.7** | **40.1** | **25.3** | **74.1** | **42.3** | **26.9** |

Table 5: Clustering performance (%) on five fine-grained datasets. The best results are in **bold**.

| Dataset | Aircraft | | | Food | | | Flowers | | | Pets | | | Cars | | |
|---|---|---|---|---|---|---|---|---|---|---|---|---|---|---|---|
| Metrics | NMI | ACC | ARI | NMI | ACC | ARI | NMI | ACC | ARI | NMI | ACC | ARI | NMI | ACC | ARI |
| TAC (KMeans) | 47.7 | 20.1 | 10.4 | 69.4 | 59.2 | 43.9 | 86.0 | 66.9 | 58.5 | 80.4 | 66.9 | 59.2 | 64.7 | 33.2 | 21.7 |
| TAC (SC) | 47.1 | 20.3 | 10.0 | 68.9 | 59.0 | 43.4 | 86.0 | 67.0 | 59.5 | 79.9 | 66.0 | 58.5 | 65.0 | 33.9 | 22.0 |
| GradNorm | 50.3 | 24.0 | 13.1 | **75.0** | **67.8** | **52.4** | 86.7 | **70.8** | **64.2** | 81.5 | **72.0** | 62.8 | **68.3** | 38.0 | **26.6** |
| Ours | **51.9** | **24.2** | **14.2** | 72.1 | 61.8 | 47.2 | **88.3** | 69.4 | 61.0 | 84.9 | **72.0** | **64.0** | 67.7 | **39.0** | 26.4 |

provement of text semantic embeddings. Differently, Li et al. (2024) and Peng et al. (2025b;c) focus on leveraging textual semantics to enhance the feature discriminability by either simply concentrating textual and visual features or its proposed cross-modal mutual distillation strategy

*Due to space limitation, more related work to this paper is discussed in Appendix A.*

## 6 CONCLUSION

In this paper, we advance spectral clustering into the multi-modal era by presenting Neural Tangent Kernel Spectral Clustering, which anchors the NTK with positive textual semantics to couple visual similarity and semantic overlap in defining affinities. This design sharpens block-diagonal structures by amplifying intra-cluster connections and suppressing cross-cluster noise, while our proposed regularized affinity diffusion further enhances robustness through adaptive ensembling of prompt-specific affinities. Empirically, our method achieves strong performance compared to competitive baselines on 16 datasets, which echoes our theoretical insights. Besides, extensive ablations provide further understandings of our proposed method. We hope this work could serve as a catalyst, motivating future studies on spectral clustering with vision-language representations, which is believable to be a promising direction for methodology improvement and real-world application.

## ACKNOWLEDGMENT

This work is supported by the Australian Research Council Discovery Early Career Researcher Award (DE250100363), Australian Future Fellowship (FT230100121), Australian Laureate Fellowship (FL190100149) and Discovery Project (DP250100463).

## ETHICS STATEMENT

Our study relies solely on publicly available datasets and models. No private or personally identifiable information was used. The work aims to advance the scientific understanding of spectral clustering while upholding principles of transparency, fairness, and responsible research.

## REPRODUCIBILITY STATEMENT

All the pre-trained CLIP-based models used in this paper are publicly accessible. We provide detailed proofs in the appendix and source codes in the supplementary materials.

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

## A MORE RELATED WORK

**Deep Clustering.** Due to its ability to reveal the inherent semantic structure underlying the data without requiring laborious and trivial data labeling work, clustering has been shown to benefit downstream tasks (Zhang et al., 2024; Peng et al., 2020; 2024a; Chen et al., 2022; Peng et al., 2025d; 2024b; 2025a) in computer vision. The popularity of deep image clustering can be attributed to the fact that distributional assumptions in classic clustering methods, e.g., compactness (Ester et al., 1996), connectivity (Wang et al., 2020; Zhu et al., 2025), sparsity (Elhamifar & Vidal, 2013; Zhu et al., 2021) and low rankness (Liu et al., 2012; Zhu & Peng, 2020), can not be necessarily conformed by high-dimensional structural RGB images. To exploit the powerful representative ability of deep neural networks in an unsupervised manner, the earliest attempts seek self-supervision signals by considering image reconstruction (Ghasedi Dizaji et al., 2017; Peng et al., 2016; Xie et al., 2016; Jiang et al., 2016; Mukherjee et al., 2019; Radford et al., 2015; Peng & Zhu, 2021) and mutual information maximization (Hu et al., 2017; Ji et al., 2019) as proxy tasks. Despite remarkable progresses, the learned representations may not be discriminative enough to capture the semantic similarity between images. More recently, the advance in self-supervised representation learning have led to major breakthroughs in deep image clustering. On the one hand, IDFD (Tao et al., 2020) proposes to perform both instance discrimination and feature de-correlation while MICE (Tsai et al., 2020) proposes a unified latent mixture model based on contrastive learning to tackle the clustering task. On the other hand, CC (Li et al., 2021b) and its followers TCC (Shen et al., 2021) perform contrastive learning at both instance and cluster levels. Differently, ProPos (Huang et al., 2022) performs non-contrastive learning on the instance level and contrastive learning on the cluster level, which results in enjoying the strengths of both worlds.

**Vision-language Models (VLMs).** Pretraining on large-scale image–text pairs has made VLMs a standard backbone for multi-modal transfer. Regarding the type of architectures, existing VLMs can be divided into two categories: (i) *single-stream* models that process concatenated visual and textual features into one transformer, such as VisualBERT (Li et al., 2019), ViLT (Kim et al., 2021), and (ii) *dual-stream* models that keep visual and text encoders seperate while learning cross-modal alignment through contrastive pairing, e.g., CLIP (Radford et al., 2021), ALIGN (Jia et al., 2021), SigLIP (Yao et al., 2021) and FILIP (Yao et al., 2021). More importantly, CLIP based models are widely adopted and has motivated a series of follow-ups aimed to improve data efficiency and better adaptation on downstream tasks (Li et al., 2021a; Zhang et al., 2021a; Zhou et al., 2022; Liao et al., 2026). This paper uses CLIP as the pre-trained model, but our method can be generally applicable to contrastive models that promote vision-language alignment.

**Neural Tangent Kernel.** Neural tangent kernel (NTK) (Golikov et al., 2022) is a kernel that reveals the connections between infinitely wide neural networks trained by gradient descent and kernel methods. NTK enables the study of neural networks using theoretical tools from the perspective of kernel methods. There have been several studies that have explored the properties of NTK: Jacot et al. (2018) propose the concept of NTK and showed that it could be used to explain the generalization of neural networks. Lee et al. (2019) expand on this work and demonstrated that the dynamics of training wide but finite-width NNs with gradient descent can be approximated by a linear model obtained from the first-order Taylor expansion of that network around its initialization. This paper, rather than exploring the interpretability of infinite width neural networks, explores empirical (i.e., finite width) NTK to construct a text-informed affinity matrix for spectral clustering.

## B DERIVATION OF EQ. (11)

We consider the following optimization problem

$$\min_{\boldsymbol{\beta},\hat{\mathbf{A}}} \sum_{b=1}^{B} \boldsymbol{\beta}[b] \cdot \ell(\hat{\mathbf{A}}, \mathbf{A}_{\text{NTK}}^{(b)}) + \mu\|\hat{\mathbf{A}} - \mathbf{E}\|_F^2 + \frac{\lambda}{2}\|\boldsymbol{\beta}\|_2^2 \quad \text{s.t.} \quad 0 \leq \boldsymbol{\beta}[b] \leq 1 \text{ and } \sum_{b=1}^{B} \boldsymbol{\beta}[b] = 1, \quad (15)$$

With fixed $\boldsymbol{\beta}$, we can simplify Eq. (15) as follows:

$$\min_{\hat{\mathbf{A}}} \sum_{b=1}^{B} \boldsymbol{\beta}[b] \cdot \ell(\hat{\mathbf{A}}, \mathbf{A}_{\text{NTK}}^{(b)}) + \mu\|\hat{\mathbf{A}} - \mathbf{E}\|_F^2, \quad (16)$$

Table 6: Spectral clustering performance (%) using different kernels for affinity measurement based on either CLIP image features or TAC image-text-concentrated features. The best results are highlighted in **bold**. † indicates our reproduced results.

| Dataset | STL-10 | | | CIFAR-10 | | | CIFAR-20 | | | ImageNet-10 | | | ImageNet-Dogs | | | DTD | | | UCF101 | | | ImageNet-1K | | | Average | | |
|---|---|---|---|---|---|---|---|---|---|---|---|---|---|---|---|---|---|---|---|---|---|---|---|---|---|---|---|
| Metrics | NMI | ACC | ARI | NMI | ACC | ARI | NMI | ACC | ARI | NMI | ACC | ARI | NMI | ACC | ARI | NMI | ACC | ARI | NMI | ACC | ARI | NMI | ACC | ARI | NMI | ACC | ARI |
| **Using CLIP image features** | | | | | | | | | | | | | | | | | | | | | | | | | | | |
| KMeans | 90.1 | 93.8 | 92.4 | 70.3 | 74.2 | 61.6 | 49.9 | 45.5 | 28.3 | 96.9 | 98.2 | 96.1 | 39.8 | 38.1 | 20.1 | 57.3 | 42.6 | 27.4 | 79.5 | 58.2 | 47.6 | 72.3 | 38.9 | 27.1 | 69.5 | 61.2 | 50.1 |
| KMeans† | 91.7 | 93.3 | 89.1 | 70.3 | 74.2 | 61.6 | 49.5 | 42.9 | 27.3 | 95.1 | 95.4 | 94.6 | 70.9 | 60.9 | 54.7 | 58.5 | 44.6 | 28.5 | 80.8 | 59.7 | 50.8 | 72.2 | 38.3 | 26.6 | 73.6 | 63.7 | 54.2 |
| Linear | 85.1 | 85.1 | 79.0 | 64.0 | 68.3 | 55.8 | 44.0 | 41.2 | 29.0 | 96.6 | 96.0 | 95.8 | 67.9 | 65.1 | 53.3 | 57.6 | 47.1 | 32.3 | 80.8 | 62.0 | 54.1 | 70.6 | 37.1 | 28.3 | 70.8 | 62.7 | 53.5 |
| Polynomial | 85.1 | 85.3 | 79.2 | 64.7 | 69.3 | 56.6 | 44.5 | 42.2 | 29.6 | 96.1 | 95.4 | 94.9 | 68.7 | 66.8 | 55.7 | 57.3 | 45.4 | 31.5 | 81.2 | 62.9 | 55.4 | 70.3 | 38.4 | 29.5 | 71.0 | 63.2 | 54.1 |
| RBF | 88.1 | 87.8 | 82.4 | 65.4 | 69.2 | 57.1 | 45.2 | 42.7 | 30.1 | 96.4 | 96.8 | 95.6 | 72.8 | 70.6 | 56.9 | 57.9 | 46.7 | 31.8 | 81.7 | 63.3 | 55.1 | 73.9 | 40.0 | 29.8 | 72.7 | 64.6 | 54.9 |
| Exponential | 83.6 | 87.8 | 80.2 | 66.6 | 70.0 | 58.3 | 45.4 | 43.3 | 30.3 | 96.3 | 94.4 | 94.1 | 70.3 | 68.5 | 54.0 | 58.9 | 46.3 | 33.3 | 81.4 | 62.4 | 54.8 | 72.9 | 39.3 | 29.3 | 71.9 | 64.0 | 54.3 |
| Laplacian | 83.7 | 87.4 | 79.3 | 67.0 | 69.6 | 58.5 | 45.4 | 43.2 | 30.7 | 96.8 | 96.6 | 94.0 | 70.6 | 69.2 | 53.9 | 59.7 | 46.1 | 34.3 | 81.2 | 62.6 | 54.7 | 69.7 | 36.1 | 27.3 | 71.8 | 63.9 | 54.1 |
| Sigmoid | 85.1 | 85.1 | 79.0 | 64.0 | 68.3 | 55.8 | 43.7 | 41.1 | 28.8 | 96.5 | 96.0 | 94.8 | 67.9 | 65.1 | 53.7 | 57.1 | 46.2 | 31.8 | 80.7 | 61.5 | 53.6 | 68.1 | 36.2 | 27.8 | 70.4 | 62.4 | 53.2 |
| **Using TAC image-text-concentrated features** | | | | | | | | | | | | | | | | | | | | | | | | | | | |
| KMeans | 92.3 | 94.5 | 89.5 | 80.8 | 90.1 | 79.8 | 60.7 | 55.8 | 42.7 | 97.5 | 98.6 | 97.0 | 75.1 | 75.1 | 63.6 | 60.1 | 45.9 | 29.0 | 81.6 | 61.3 | 52.4 | 77.8 | 48.9 | 36.4 | 78.2 | 71.3 | 61.3 |
| Linear | 90.0 | 90.8 | 91.1 | 79.8 | 88.8 | 79.3 | 56.1 | 54.0 | 29.9 | 96.8 | 97.5 | 96.9 | 71.9 | 71.0 | 62.1 | 59.0 | 45.6 | 30.9 | 80.1 | 61.5 | 53.9 | 76.5 | 47.8 | 34.3 | 76.3 | 69.6 | 59.8 |
| Polynomial | 90.2 | 90.6 | 94.0 | 81.5 | 90.0 | 79.8 | 56.6 | 54.5 | 30.1 | 96.9 | 97.6 | 96.8 | 73.0 | 73.6 | 63.1 | 59.4 | 46.3 | 29.8 | 80.0 | 61.6 | 52.3 | 77.0 | 48.0 | 34.5 | 76.8 | 70.3 | 60.1 |
| RBF | 92.6 | 94.3 | 94.2 | 81.2 | 90.3 | 80.1 | 56.9 | 54.5 | 30.1 | 97.0 | 98.3 | 96.8 | 75.3 | 75.8 | 64.4 | 58.6 | 44.0 | 27.1 | 79.6 | 60.0 | 50.1 | 78.0 | 49.1 | 36.2 | 77.4 | 70.8 | 59.9 |
| Exponential | 91.5 | 94.5 | 94.0 | 81.6 | 90.5 | 80.5 | 57.2 | 54.6 | 30.3 | 96.9 | 98.0 | 95.0 | 74.5 | 73.6 | 62.5 | 59.0 | 45.8 | 28.8 | 79.9 | 62.1 | 53.5 | 77.5 | 50.0 | 36.0 | 77.3 | 71.1 | 60.1 |
| Laplacian | 92.0 | 94.0 | 93.6 | 82.3 | 90.9 | 80.3 | 57.0 | 54.9 | 31.0 | 97.2 | 98.1 | 95.9 | 75.0 | 74.2 | 62.8 | 59.5 | 46.0 | 32.0 | 80.1 | 63.0 | 54.9 | 74.6 | 47.0 | 35.1 | 77.2 | 71.0 | 60.7 |
| Sigmoid | 93.6 | 94.4 | 93.1 | 80.1 | 88.5 | 76.9 | 55.7 | 52.3 | 29.0 | 96.8 | 97.5 | 95.3 | 72.2 | 70.9 | 60.1 | 58.0 | 46.2 | 32.3 | 80.5 | 61.9 | 53.2 | 72.9 | 46.6 | 35.8 | 76.2 | 69.8 | 59.5 |
| Ours | **95.8** | **98.3** | **96.3** | **83.3** | **92.0** | **83.0** | **63.3** | **59.6** | **43.5** | **97.8** | **99.2** | **98.4** | **82.4** | **84.9** | **71.4** | **61.7** | **52.0** | **33.6** | **83.0** | **67.9** | **59.4** | **79.2** | **56.3** | **39.4** | **80.8** | **76.3** | **65.6** |

where

$$\ell(\hat{\mathbf{A}}, \mathbf{A}_{\text{NTK}}^{(b)}) = \frac{1}{2} \sum_{i,j,k,l=1}^{M} a_{ij}^{(b)} a_{kl}^{(b)} \left( \frac{\hat{\mathbf{A}}[k,i]}{\sqrt{d_i^{(b)} d_k^{(b)}}} - \frac{\hat{\mathbf{A}}[l,j]}{\sqrt{d_j^{(b)} d_l^{(b)}}} \right) \tag{17}$$

with $a_{ij}^{(b)} = \mathbf{A}_{\text{NTK}}^{(b)}[i,j]$ and $d_i^{(b)} = \sum_{j=1}^{M} \mathbf{A}_{\text{NTK}}^{(b)}[i,j]$.

Let us define

$$\mathbb{W}^{(b)} = \mathbf{A}_{\text{NTK}}^{(b)} \otimes \mathbf{A}_{\text{NTK}}^{(b)} \in \mathbb{R}^{M^2 \times M^2}$$

$$\mathbb{T}^{(b)} = \mathbf{D}_{\text{NTK}}^{(b)} \otimes \mathbf{D}_{\text{NTK}}^{(b)} \in \mathbb{R}^{M^2 \times M^2}$$

with $\mathbf{D}_{\text{NTK}}^{(b)}$ as a diagonal matrix of $\mathbf{D}_{\text{NTK}}^{(b)}[i,i] = \sum_{j=1}^{M} \mathbf{A}_{\text{NTK}}^{(b)}[i,j]$, and

$$\mathbb{S}^{(b)} = \mathbf{S}_{\text{NTK}}^{(b)} \otimes \mathbf{S}_{\text{NTK}}^{(b)} \in \mathbb{R}^{M^2 \times M^2}$$

with $\mathbf{S}_{\text{NTK}}^{(b)}$ as the row-normalized $\mathbf{A}_{\text{NTK}}^{(b)}$, i.e., $\mathbf{S}_{\text{NTK}}^{(b)} = \left( \mathbf{D}_{\text{NTK}}^{(b)} \right)^{-1/2} \mathbf{A}_{\text{NTK}}^{(b)} \left( \mathbf{D}_{\text{NTK}}^{(b)} \right)^{-1/2}$

One can easily check that $\|\hat{\mathbf{A}} - \mathbf{E}\|_F^2 = \|vec(\hat{\mathbf{A}}) - vec(\mathbf{E})\|_2^2$.

By introducing two identical coordinate transformations: $\varepsilon \equiv M(i-1) + k$ and $\delta \equiv M(j-1) + l$, we have the following:

$$\begin{aligned}
\ell(\hat{\mathbf{A}}, \mathbf{A}_{\text{NTK}}^{(b)}) &= \frac{1}{2} \sum_{\varepsilon,\delta=1}^{M^2} w_{\varepsilon,\delta}^{(b)} \left( \frac{\hat{\mathbf{a}}[\varepsilon]}{\sqrt{t_{\varepsilon\varepsilon}^{(b)}}} - \frac{\hat{\mathbf{a}}[\delta]}{\sqrt{t_{\delta\delta}^{(b)}}} \right)^2 \\
&= \sum_{\varepsilon,\delta=1}^{M^2} w_{\varepsilon,\delta}^{(b)} \frac{\hat{\mathbf{a}}[\varepsilon]^2}{t_{\varepsilon\varepsilon}^{(b)}} - \sum_{\varepsilon,\delta=1}^{M^2} \hat{\mathbf{a}}[\varepsilon] \frac{w_{\varepsilon,\delta}^{(b)}}{\sqrt{t_{\varepsilon\varepsilon}^{(b)} t_{\delta\delta}^{(b)}}} \hat{\mathbf{a}}[\delta] \\
&= \sum_{\varepsilon=1}^{M^2} \hat{\mathbf{a}}[\varepsilon]^2 - \hat{\mathbf{a}}^{\top} \mathbb{T}^{(b)-1/2} \mathbb{W}^{(b)} \mathbb{T}^{(b)-1/2} \hat{\mathbf{a}} \\
&= \hat{\mathbf{a}}^{\top} \left( \mathbf{I}_{M^2} - \mathbb{T}^{(b)-1/2} \mathbb{W}^{(b)} \mathbb{T}^{(b)-1/2} \right) \hat{\mathbf{a}} \\
&= \hat{\mathbf{a}}^{\top} \left( \mathbf{I}_{M^2} - \mathbb{S}^{(b)} \right) \hat{\mathbf{a}},
\end{aligned} \tag{18}$$

where $\hat{\mathbf{a}} = vec(\hat{\mathbf{A}})$, $w_{\varepsilon,\delta}^{(b)} = \mathbb{W}^{(b)}[\varepsilon,\delta]$ and $t_{\delta\delta}^{(b)} = \mathbb{T}^{(b)}[\delta,\delta]$.

The following three facts are applied for the derivation of Eq. (18):

1. $\mathbb{W}^{(b)}$ is symmetric since $\mathbf{W}^{(b)}$ is symmetric.

2. $\mathbb{T}^{(b)}[\delta, \delta] = \sum_{\varepsilon=1}^{M^2} \mathbb{W}^{(b)}[\delta, \varepsilon]$ since

$$
\begin{aligned}
\mathbb{T}^{(b)}[\delta, \delta] &= \mathbb{D}_{NTK}^{(b)}[i, i]\mathbb{D}_{NTK}^{(b)}[k, k] \\
&= \left( \sum_{j=1}^{M} \mathbf{A}_{\text{NTK}}^{(b)}[i, j] \right) \left( \sum_{l=1}^{M} \mathbf{A}_{\text{NTK}}^{(b)}[k, l] \right) \\
&= \sum_{j=1}^{M} \sum_{l=1}^{M} \mathbf{A}_{\text{NTK}}^{(b)}[i, j]\mathbf{A}_{\text{NTK}}^{(b)}[k, l] = \sum_{\varepsilon=1}^{M^2} \mathbb{W}^{(b)}[\delta, \varepsilon]
\end{aligned}
\tag{19}
$$

3. $\mathbb{S}^{(b)} = \mathbb{T}^{(b)^{-1/2}}\mathbb{W}^{(b)}\mathbb{T}^{(b)^{-1/2}}$ since

$$
\begin{aligned}
\mathbb{S}^{(b)}[\delta, \varepsilon] &= \mathbf{S}_{NTK}^{(b)}[i, j]\mathbf{S}_{NTK}^{(b)}[k, l] \\
&= \mathbf{D}_{NTK}^{(b)}[i, i]^{-0.5}\mathbf{A}_{NTK}^{(b)}[i, j]\mathbf{D}_{NTK}^{(b)}[j, j]^{-0.5}\mathbf{D}_{NTK}^{(b)}[k, k]^{-0.5}\mathbf{A}_{NTK}^{(b)}[k, l]\mathbf{D}_{NTK}^{(b)}[l, l]^{-0.5} \\
&= \mathbf{D}_{NTK}^{(b)}[i, i]^{-0.5}\mathbf{D}_{NTK}^{(b)}[k, k]^{-0.5}\mathbf{A}_{NTK}^{(b)}[i, j]\mathbf{A}_{NTK}^{(b)}[k, l]\mathbf{D}_{NTK}^{(b)}[j, j]^{-0.5}\mathbf{D}_{NTK}^{(b)}[l, l]^{-0.5} \\
&= \mathbb{T}^{(b)}[\delta, \delta]^{-0.5}\mathbb{W}^{(b)}[\delta, \varepsilon]\mathbb{T}^{(b)}[\varepsilon, \varepsilon]^{-0.5}
\end{aligned}
\tag{20}
$$

In summary, the objective function in Eq. (15) can be rewritten as follows:

$$
J = \sum_{b=1}^{B} \boldsymbol{\beta}[b] \cdot \hat{\mathbf{a}}^{\top} \left( \mathbf{I}_{M^2} - \mathbb{S}^{(b)} \right) \hat{\mathbf{a}} + \mu \|\hat{\mathbf{a}} - vec(\mathbf{E})\|_2^2
\tag{21}
$$

## C    DERIVATION OF EQ. (12)

By taking the partial derivative of $J$ in Eq. (21) with regard to $\hat{\mathbf{a}}$, we have the following:

$$
\begin{aligned}
\frac{\partial J}{\partial \hat{\mathbf{a}}} &= \sum_{b=1}^{B} \boldsymbol{\beta}[b] \cdot \frac{\partial}{\partial \hat{\mathbf{a}}} \left\{ \hat{\mathbf{a}}^{\top} \left( \mathbf{I}_{M^2} - \mathbb{S}^{(b)} \right) \hat{\mathbf{a}} \right\} + \mu \frac{\partial}{\partial \hat{\mathbf{a}}} \left\{ \|\hat{\mathbf{a}} - vec(\mathbf{E})\|_2^2 \right\} \\
&= \sum_{b=1}^{B} \boldsymbol{\beta}[b] \cdot \left[ 2 \left( \mathbf{I}_{M^2} - \mathbb{S}^{(b)} \right) \hat{\mathbf{a}} \right] + 2\mu \left( \hat{\mathbf{a}} - vec(\mathbf{E}) \right)
\end{aligned}
\tag{22}
$$

By setting $\partial J/\partial \hat{\mathbf{a}} = 0$, we have the following:

$$
vec(\hat{\mathbf{A}}) = \hat{\mathbf{a}} = \frac{\mu}{\mu + 1} \cdot \left( \mathbf{I}_{M^2} - \sum_{b=1}^{B} \frac{\boldsymbol{\beta}[b]}{\mu + 1}\mathbb{S}^{(b)} \right)^{-1} vec(\mathbf{E}).
\tag{23}
$$

Applying $vec^{-1}(\cdot)$ to both sides of Eq. (23) results in the following:

$$
\hat{\mathbf{A}} = \frac{\mu}{\mu + 1} \cdot vec^{-1} \left( \left( \mathbf{I}_{M^2} - \sum_{b=1}^{B} \frac{\boldsymbol{\beta}[b]}{\mu + 1}\mathbb{S}^{(b)} \right)^{-1} vec(\mathbf{E}) \right),
\tag{24}
$$

## D    CONVERGENCE OF EQ. (13) TO EQ. (12)

**Lemma 1.** *Let $\mathbf{A} \in \mathbb{R}^{n \times n}$, the spectral radius of $\mathbf{A}$ is denoted as $\rho(\mathbf{A}) = \max\{|\lambda|, \lambda \in \sigma(\mathbf{A})\}$, where $\sigma(\mathbf{A})$ is the spectrum of $\mathbf{A}$ that represents the set of all the eigenvalues. Let $\|\cdot\|$ be a matrix norm on $\mathbb{R}^{n \times n}$, given a square matrix $\mathbf{A} \in \mathbb{R}^{n \times n}$, $\lambda$ is an arbitrary eigenvalue of $\mathbf{A}$, then we have $|\lambda| \le \rho(\mathbf{A}) \le \|\mathbf{A}\|$.*

**Lemma 2.** *Let $\mathbf{A} \in \mathbb{R}^{m \times m}$, $\mathbf{B} \in \mathbb{R}^{n \times n}$, denote $\{\lambda_i, \boldsymbol{x}_i\}_{i=1}^{m}$ and $\{\mu_i, \boldsymbol{y}_i\}_{i=1}^{n}$ as the eigen-pairs of $\mathbf{A}$ and $\mathbf{B}$ respectively. The set of $mn$ eigen-pairs of $\mathbf{A} \otimes \mathbf{B}$ is given by:*

$$
\{\lambda_i\mu_j, \boldsymbol{x}_i \otimes \boldsymbol{y}_j\}_{i=1,\dots,m, \ j=1,\dots n}.
$$

**Lemma 3.** *Let $\mathbf{A} \in \mathbb{R}^{m \times n}$, $\mathbf{X} \in \mathbb{R}^{n \times p}$ and $\mathbf{B} \in \mathbb{R}^{p \times q}$ respectively, then*

$$
vec(\mathbf{A}\mathbf{X}\mathbf{B}) = (\mathbf{B}^{\top} \otimes \mathbf{A})vec(\mathbf{X}).
$$

**Lemma 4.** *Let $A \in \mathbb{R}^{n \times n}$, then $\lim_{k \to \infty} A^k = 0$ if and only if $\rho(A) < 1$.*

**Lemma 5.** *Given a matrix $A \in \mathbb{R}^{n \times n}$ and $\rho(A) < 1$, the Neumann series $I_n + A + A^2 + \cdots$ converges to $(I_n - A)^{-1}$.*

To get started, we first consider the matrix $\left(\mathbf{D}_{\text{NTK}}^{(b)}\right)^{-1} \mathbf{A}_{\text{NTK}}^{(b)}$, whose induced $l_\infty$-norm is equal to 1, i.e., $\|\left(\mathbf{D}_{\text{NTK}}^{(b)}\right)^{-1} \mathbf{A}_{\text{NTK}}^{(b)}\|_\infty = 1$, since the $i$-th diagonal element in matrix $\mathbf{D}_{\text{NTK}}^{(b)}$ equal to the summation of the corresponding $i$-th row in matrix $\mathbf{A}_{\text{NTK}}^{(b)}$. Lemma 1 gives that $\rho\left(\left(\mathbf{D}_{\text{NTK}}^{(b)}\right)^{-1} \mathbf{A}_{\text{NTK}}^{(b)}\right) \leq 1$.

As for the matrix $\mathbf{S}_{\text{NTK}}^{(b)} = \left(\mathbf{D}_{\text{NTK}}^{(b)}\right)^{-1/2} \mathbf{A}_{\text{NTK}}^{(b)} \left(\mathbf{D}_{\text{NTK}}^{(b)}\right)^{-1/2}$ we are concerned about, since we can rewrite it as $\mathbf{S}_{\text{NTK}}^{(b)} = \left(\mathbf{D}_{\text{NTK}}^{(b)}\right)^{1/2} \left(\mathbf{D}_{\text{NTK}}^{(b)}\right)^{-1} \mathbf{A}_{\text{NTK}}^{(b)} \left(\mathbf{D}_{\text{NTK}}^{(b)}\right)^{-1/2}$, thus it is similar to $\left(\mathbf{D}_{\text{NTK}}^{(b)}\right)^{-1} \mathbf{A}_{\text{NTK}}^{(b)}$. This implies that the two matrices share the same eigenvalues, such that $\rho\left(\mathbf{S}_{\text{NTK}}^{(b)}\right) \leq 1$. By applying Lemma 2, we can conclude that both the spectral radius of the Kronecker product $\mathbb{S}^{(b)} = \mathbf{S}_{\text{NTK}}^{(b)} \otimes \mathbf{S}_{\text{NTK}}^{(b)}$ is no larger than 1, i.e., $\rho\left(\mathbb{S}^{(b)}\right) \leq 1$.

By applying Lemma 3, Eq. (13) can be vectorized as the following:

$$
\begin{aligned}
\hat{\mathbf{a}}^{(p+1)} &= \sum_{b=1}^{B} \frac{\boldsymbol{\beta}[b]}{\mu + 1} \left(\mathbf{S}^{(b)}\right)^\top \otimes \mathbf{S}^{(b)} \hat{\mathbf{a}}^{(p)} + \frac{\mu}{\mu + 1} vec(\mathbf{E}) \\
&= \sum_{b=1}^{B} \frac{\boldsymbol{\beta}[b]}{\mu + 1} \mathbf{S}^{(b)} \otimes \mathbf{S}^{(b)} \hat{\mathbf{a}}^{(p)} + \frac{\mu}{\mu + 1} vec(\mathbf{E}) \\
&= \sum_{b=1}^{B} \frac{\boldsymbol{\beta}[b]}{\mu + 1} \mathbb{S}^{(b)} \hat{\mathbf{a}}^{(p)} + \frac{\mu}{\mu + 1} vec(\mathbf{E}) \\
&= \sum_{b=1}^{B} \frac{\boldsymbol{\beta}[b]}{\mu + 1} \mathbb{S}^{(b)} \left(\sum_{b=1}^{B} \frac{\boldsymbol{\beta}[b]}{\mu + 1} \mathbb{S}^{(b)} \hat{\mathbf{a}}^{(p)} + \frac{\mu}{\mu + 1} vec(\mathbf{E})\right) + \frac{\mu}{\mu + 1} vec(\mathbf{E}) \\
&= \left(\sum_{b=1}^{B} \frac{\boldsymbol{\beta}[b]}{\mu + 1} \mathbb{S}^{(b)}\right)^p \hat{\mathbf{a}}^{(1)} + \frac{\mu}{\mu + 1} \sum_{i=0}^{p-1} \left(\sum_{b=1}^{B} \frac{\boldsymbol{\beta}[b]}{\mu + 1} \mathbb{S}^{(b)}\right)^i vec(\mathbf{E})
\end{aligned}
\tag{25}
$$

where the second step is derived based on the fact that $\mathbf{S}^{(b)}$ is symmetric.

Since we have already proved that $\rho\left(\mathbb{S}^{(b)}\right) \leq 1$, we have the spectral radius of $\sum_{b=1}^{B} \frac{\boldsymbol{\beta}[b]}{\mu+1} \mathbb{S}^{(b)}$ to be upper-bounded by $\sum_{b=1}^{B} \frac{\boldsymbol{\beta}[b]}{\mu+1}$. Moreover, since $\mu > 0$, we have

$$
\rho\left(\sum_{b=1}^{B} \frac{\boldsymbol{\beta}[b]}{\mu + 1} \mathbb{S}^{(b)}\right) \leq \sum_{b=1}^{B} \frac{\boldsymbol{\beta}[b]}{\mu + 1} = \frac{\sum_{b=1}^{B} \boldsymbol{\beta}[b]}{\mu + 1} = \frac{1}{\mu + 1} < 1
\tag{26}
$$

By taking advantage of Lemma 4 and 5, we can easily demonstrate that the following two expressions hold true:

$$
\lim_{p \to \infty} \left(\sum_{b=1}^{B} \frac{\boldsymbol{\beta}[b]}{\mu + 1} \mathbb{S}^{(b)}\right)^p = 0,
\tag{27}
$$

$$
\sum_{i=0}^{p-1} \left(\sum_{b=1}^{B} \frac{\boldsymbol{\beta}[b]}{\mu + 1} \mathbb{S}^{(b)}\right)^i = \left(\mathbf{I}_{M^2} - \sum_{v=1}^{m} \frac{\boldsymbol{\beta}[b]}{\mu + 1} \mathbb{S}^{(b)}\right)^{-1}.
\tag{28}
$$

Therefore, the iterative sequence of $\hat{\mathbf{a}}^{(p+1)}$ asymptotically approaches a stable solution, converging to:

$$
\lim_{p \to \infty} \hat{\mathbf{a}}^{(p+1)} = \frac{\mu}{\mu + 1} \left(\mathbf{I}_{M^2} - \sum_{v=1}^{m} \frac{\boldsymbol{\beta}[b]}{\mu + 1} \mathbb{S}^{(b)}\right)^{-1} vec(\mathbf{E}).
\tag{29}
$$

By performing the inverse operator $vec^{-1}(\cdot)$ on the right side of Eq. (29), we have the following

$$\hat{\mathbf{A}}^* = \frac{\mu}{\mu+1} \cdot vec^{-1}\left(\left(\mathbf{I}_{M^2} - \sum_{b=1}^{B} \frac{\boldsymbol{\beta}[b]}{\mu+1}\mathbb{S}^{(b)}\right)^{-1} vec(\mathbf{E})\right), \tag{30}$$

## E   OPTIMIZE $\beta$ WITH FIXED $\hat{\mathbf{A}}$

Again, we consider the following optimization problem

$$\min_{\boldsymbol{\beta}, \hat{\mathbf{A}}} \sum_{b=1}^{B} \boldsymbol{\beta}[b] \cdot \ell(\hat{\mathbf{A}}, \mathbf{A}_{\text{NTK}}^{(b)}) + \mu\|\hat{\mathbf{A}} - \mathbf{E}\|_F^2 + \frac{\lambda}{2}\|\boldsymbol{\beta}\|_2^2 \quad \text{s.t.} \quad 0 \le \boldsymbol{\beta}[b] \le 1 \text{ and } \sum_{b=1}^{B} \boldsymbol{\beta}[b] = 1, \tag{31}$$

When $\boldsymbol{F}$ is fixed, the objective value of $\ell(\hat{\mathbf{A}}, \mathbf{A}_{\text{NTK}}^{(b)})$ for each adjacency matrix $\mathbf{A}_{\text{NTK}}^{(b)}$ in Eq. (31) can be directly computed. As a result, the optimization of $\boldsymbol{\beta}$ reduces to solving the following problem:

$$\min_{\boldsymbol{\beta}} \sum_{b=1}^{B} \boldsymbol{\beta}[b] \cdot \ell(\hat{\mathbf{A}}, \mathbf{A}_{\text{NTK}}^{(b)}) + \frac{\lambda}{2}\|\boldsymbol{\beta}\|_2^2, \quad s.t. \quad 0 \le \boldsymbol{\beta}[b] \le 1 \text{ and } \sum_{b=1}^{B} \boldsymbol{\beta}[b] = 1. \tag{32}$$

Specifically, the objective function in Eq. (32) takes the form of a Lasso optimization problem, which can be solved by utilizing the coordinate descent method.

In each iteration of the coordinate descent, two elements $\boldsymbol{\beta}[i]$ and $\boldsymbol{\beta}[j]$ are selected to be updated, while the others are fixed. Taking into account the Lagrange function for the constraint $\sum_{b=1}^{B} \boldsymbol{\beta}[b] = 1$, we have the following updating scheme:

$$\boldsymbol{\beta}^*[i] \leftarrow \frac{\lambda(\boldsymbol{\beta}[i] + \boldsymbol{\beta}[j]) + (H^{(j)} - H^{(i)})}{2\lambda} \tag{33}$$

$$\boldsymbol{\beta}^*[j] \leftarrow \boldsymbol{\beta}[i] + \boldsymbol{\beta}[j] - \boldsymbol{\beta}^*[i], \tag{34}$$

where $H^{(b)} = \ell(\hat{\mathbf{A}}, \mathbf{A}_{\text{NTK}}^{(b)})$. To avoid the obtained $\boldsymbol{\beta}^*[i]$ and $\boldsymbol{\beta}^*[j]$ to violate the constraint $0 \le \boldsymbol{\beta}[b]$, we set $\boldsymbol{\beta}^*[i] = 0$ if $\lambda(\boldsymbol{\beta}[i] + \boldsymbol{\beta}[j]) + (H^{(j)} - H^{(i)}) < 0$, and $\boldsymbol{\beta}^*[j] = 0$ otherwise.

However, this strategy requires **multiple** iterations since only a pair of elements of $\boldsymbol{\beta}$ can be updated together. To address this issue, we propose a more efficient solution that allows updating all elements of $\boldsymbol{\beta}$ simultaneously, explicitly eliminating the need for iteration.

By taking advantage of the coordinate descent method, we can filter out the valid elements that are not governed by the boundary constraints, formally denoting the valid index set as $\mathcal{B}$. Consequently, the inequality constraints of $0 \le \boldsymbol{\beta}[i] \le 1$ are slack to the weight set $\{\boldsymbol{\beta}[b]\}_{b\in\mathcal{B}}$ and the optimization problem can be directly solved.

In particular, by introducing a Lagrangian multiplier $\eta$, the Lagrangian function $\mathcal{L}(\boldsymbol{\beta}, \eta)$ can be formally defined as:

$$\mathcal{L}(\boldsymbol{\beta}, \eta) = \sum_{b\in\mathcal{B}} \boldsymbol{\beta}[b] \cdot H^{(b)} + \frac{\lambda}{2}\|\boldsymbol{\beta}\|_2^2 + \eta(1 - \sum_{b\in\mathcal{B}} \boldsymbol{\beta}[b]). \tag{35}$$

The corresponding Karush-Kuhn-Tucker (KKT) conditions can then be formulated as:

$$\begin{cases} \nabla_{\boldsymbol{\beta}[b]}\mathcal{L}(\boldsymbol{\beta}, \eta) = \dfrac{\partial\mathcal{L}(\boldsymbol{\beta}, \eta)}{\partial\boldsymbol{\beta}[b]} = H^{(b)} + \lambda\boldsymbol{\beta}[b] - \eta = 0, \quad b \in \mathcal{B}, \\ \nabla_{\eta}\mathcal{L}(\beta, \eta) = \dfrac{\partial\mathcal{L}(\boldsymbol{\beta}, \eta)}{\partial\eta} = 1 - \sum_{b\in\mathcal{B}} \boldsymbol{\beta}[b] = 0. \end{cases} \tag{36}$$

Note that we have already taken the equation constraint $\sum_{b\in\mathcal{B}} \boldsymbol{\beta}[b] = 1$ into consideration when deriving the representation of $\nabla_{\boldsymbol{\beta}[b]}\mathcal{L}(\boldsymbol{\beta}, \eta)$. The optimal result can be obtained by solving the $|\mathcal{B}| + 1$ equations. By summing up all the $\nabla_{\boldsymbol{\beta}[b]}\mathcal{L}(\boldsymbol{\beta}, \eta)$ along $b$ within $\mathcal{B}$, the Lagrangian multiplier $\eta$ can be obtained as:

$$\eta = \frac{\sum_{b\in\mathcal{B}} H^{(b)} + \lambda}{|\mathcal{B}|}. \tag{37}$$

---

**Algorithm 1:** Affinity Ensembling via Regularized Affinity Diffusion

---

**Input:** Affinity matrices $\{\mathbf{A}_{\text{NTK}}^{(b)}\}_{b=1}^{B}$, the reference matrix $\mathbf{E}$, max number of iterations $maxiter$, weighting parameter $\mu$ and $\lambda$

**Output:** The ensembled affinity matrix $\hat{\mathbf{A}}$

1. initialize $t = 0$, $\hat{\mathbf{A}} = \mathbf{E}$, and $\boldsymbol{\beta}[b] = 1/B$ for each $b = 1, \ldots, B$

2. **repeat**

3.    update $\hat{\mathbf{A}}$ with fixed $\boldsymbol{\beta}$ following Eq. (13)

4.    compute $H^{(b)} = \ell(\hat{\mathbf{A}}, \mathbf{A}_{\text{NTK}}^{(b)})$ for each $b = 1, \ldots, B$

5.    set $\boldsymbol{\beta}[b] = 0$ for each $b = 1, \ldots, B$

6.    filter the valid index set $\mathcal{B}$ following Eq. (39)

7.    update $\boldsymbol{\beta}[b]$ with fixed $\hat{\mathbf{A}}$ following Eq. (40) for each $b \in \mathcal{B}$

9.    $t \leftarrow t + 1$

10. **until** convergence or $t = maxiter$

---

Therefore, by taking $\eta$ back into the KKT conditions, we can obtain the optimal solution of $\boldsymbol{\beta}^*[b]$, following:

$$\boldsymbol{\beta}^*[b] = \frac{\sum_{b' \in \mathcal{B}} H^{b'} - |\mathcal{B}| H^b + \lambda}{\lambda |\mathcal{B}|}, v \in \mathcal{I}. \tag{38}$$

Since all the element $\boldsymbol{\beta}^*[b]$ should satisfy the inequality constraint $0 \leq \boldsymbol{\beta}^*[b] \leq 1$, the above relationships provide an effective strategy to determine the valid index set $\mathcal{B}$, *i.e.*, the corresponding $H^b$ in $\mathcal{B}$ should satisfy $H^b \leq (\sum_{b' \in \mathcal{B}} H^{b'} + \lambda)/|\mathcal{B}|$. Therefore, we can develop a formalize definition of the valid index set, as follows:

$$\mathcal{B} = \left\{ v | H^v < (\sum_{b' \in \mathcal{B}} H^{b'} + \lambda)/|\mathcal{B}|, v = 1, 2, \ldots, m \right\}. \tag{39}$$

In practical implementation, we first sort all $H^b$ in descending order and then sequentially remove the indices that fail to satisfy the constraint of Eq. (39), leading to the valid set $\mathcal{B}$. The optimal result can be obtained in **a single round of iteration**, with the resulting weight vector $\boldsymbol{\beta}^*$ given by:

$$\boldsymbol{\beta}^*[b] = \begin{cases} \dfrac{\sum_{b' \in \mathcal{B}} H^{b'} - |\mathcal{B}| H^b + \lambda}{\lambda |\mathcal{B}|}, v \in \mathcal{B}, \\ 0, \quad v \in \{1, 2, \ldots, B\}/\mathcal{I}. \end{cases} \tag{40}$$

## F ABLATION ON REGULARIZED AFFINITY DIFFUSION

In this section, we conduct ablation study on our proposed Regularized Affinity Diffusion (RAD) mechanism in Section 3.2. To examine the effectiveness of RAD, we consider the following two alternatives to RAD.

The first alternative is to naively average the obtained $B$ affinity matrices $\mathbf{A}_{\text{NTK}}^{(1)}, \ldots, \mathbf{A}_{\text{NTK}}^{(B)}$, i.e.,

$$\hat{\mathbf{A}} = \frac{1}{B} \sum_{b=1}^{B} \mathbf{A}_{\text{NTK}}^{(b)}. \tag{41}$$

We refer this alternative as Ours (naive).

The second alternative, motivated by Prompt Ensembling (PE), uses Eq. (6) and Eq. (7) to construct the ensembled affinity matrix $\hat{\mathbf{A}}$ except that $\boldsymbol{\theta}_0 = vec\left(\bar{\mathbf{W}}\right) = vec\left(\frac{1}{B} \sum_{b=1}^{B} \mathbf{W}^{(b)}\right)$ where $\mathbf{W}^{(b)}$ is the CLIP features of positive nouns induced by $b$-th prompt template $\Delta^{(b)}(\cdot)$. Formally, $\hat{\mathbf{A}}$ computed

Table 7: Clustering performance (%) on eight image clustering datasets between prompt ensemble and our methods. The best results are in **bold**.

| Dataset | STL-10 | | | CIFAR-10 | | | CIFAR-20 | | | ImageNet-10 | | | ImageNet-Dogs | | | DTD | | | UCF101 | | | ImageNet-1K | | | Average | | |
|---|---|---|---|---|---|---|---|---|---|---|---|---|---|---|---|---|---|---|---|---|---|---|---|---|---|---|---|
| Metrics | NMI | ACC | ARI | NMI | ACC | ARI | NMI | ACC | ARI | NMI | ACC | ARI | NMI | ACC | ARI | NMI | ACC | ARI | NMI | ACC | ARI | NMI | ACC | ARI | NMI | ACC | ARI |
| TAC (Kmeans) | 92.3 | 94.5 | 89.5 | 80.8 | 90.1 | 79.8 | 60.7 | 55.8 | 42.7 | 97.5 | 98.6 | 97.0 | 75.1 | 75.1 | 63.6 | 60.1 | 45.9 | 29.0 | 81.6 | 61.3 | 52.4 | 77.8 | 48.9 | 36.4 | 78.3 | 71.3 | 61.3 |
| TAC (SC) | 92.6 | 94.3 | 94.2 | 81.2 | 90.3 | 80.1 | 56.9 | 54.5 | 30.1 | 97.0 | 98.3 | 96.8 | 75.3 | 75.8 | 64.4 | 58.6 | 44.0 | 27.1 | 79.6 | 60.0 | 50.1 | 78.0 | 49.1 | 36.2 | 77.4 | 70.8 | 59.9 |
| Ours (naive) | 87.0 | 91.2 | 84.0 | 71.4 | 74.7 | 56.8 | 45.1 | 42.6 | 29.9 | 86.1 | 90.4 | 80.9 | 70.5 | 69.0 | 56.1 | 56.2 | 45.6 | 30.2 | 82.2 | 64.9 | 57.9 | 77.1 | 51.0 | 35.4 | 72.0 | 66.2 | 53.9 |
| Ours (PE) | 93.1 | 97.9 | 89.6 | 82.9 | 91.3 | 82.4 | 60.9 | 55.0 | 43.4 | 97.6 | 98.9 | 97.4 | 82.3 | 81.3 | 72.3 | 60.7 | 50.6 | 32.1 | 81.5 | 66.8 | 58.9 | 79.2 | 53.3 | 38.4 | 79.8 | 74.4 | 64.3 |
| Ours (RAD) | **95.8** | **98.3** | **96.3** | **83.3** | **92.0** | **83.0** | **63.3** | **59.6** | **43.5** | **97.8** | **99.2** | **98.4** | **82.4** | **84.9** | **71.4** | **61.7** | **52.0** | **33.6** | **83.0** | **67.9** | **59.4** | **79.2** | **56.3** | **39.4** | **80.8** | **76.3** | **65.6** |

by the second alternative, called Ours (PE), can be given as follows:

$$\hat{\mathbf{A}}[i,j] = \begin{cases} \mathcal{K}_{\bar{\mathbf{W}}}\left(f_{\mathcal{X}}(\mathbf{x}_i), f_{\mathcal{X}}(\mathbf{x}_j)\right) & \text{if } f_{\mathcal{X}}(\mathbf{x}_i) \in \mathcal{N}_q(f_{\mathcal{X}}(\mathbf{x}_j), \mathcal{K}_{\bar{\mathbf{W}}}) \wedge f_{\mathcal{X}}(\mathbf{x}_j) \in \mathcal{N}_q(f_{\mathcal{X}}(\mathbf{x}_i), \mathcal{K}_{\bar{\mathbf{W}}}). \\ 0 & \text{otherwise.} \end{cases} \quad (42)$$

where

$$g_{\bar{\mathbf{W}}}\left(f_{\mathcal{X}}(\mathbf{x}_i)\right) = \log \sum_{k=1}^{N} e^{\bar{\mathbf{w}}_k^\top f_{\mathcal{X}}(\mathbf{x}_i)/\tau} \quad (43)$$

with $\bar{\mathbf{w}}_k = \frac{1}{B}\sum_{b=1}^{B} f_{\mathcal{T}}\left(\Delta^{(b)}(\hat{\mathbf{c}}_k)\right)$.

Table 7 shows that Ours (RAD) achieves the best clustering performance, which validates the effectiveness. Besides, one can also find that Ours (PE) significantly outperforms TAC (SC) and TAC (KMeans), which implies the effectiveness of our proposed NTK-induced affinity measure.

## G  BROADER IMPACT

This work proposes a new deep clustering paradigm by leveraging external knowledge. As a fundamental problem in machine learning, clustering has a wide range of applications, such as anomaly detection, person re-identification, community detection, etc. The proposed method is evaluated on public image datasets that are not at risk. However, just like any learning method, the performance of our method depends on data bias and cannot be guaranteed in more complex real-world applications. In this sense, it might bring some disturbances in decision-making and thus should be carefully used, especially in areas such as health care, autonomous vehicles, etc.

## H  USAGE CLAIM OF LLMS.

We use ChatGPT for grammar and spelling checks only, with prompt "Proofread the sentences".

## I  LIMITATION

The proposed method requires manually setting the target cluster number. In real-world applications, one may resort to other cluster number estimation methods in the lack of a cluster number prior.

## J  ABLATION STUDY ON PROMPT TEMPLATES

To investigate how much has been prompt ensembling contributing to the performance compared to using only one prompt, we report the clustering performance under single-prompt setting in Table 8.

Even with a single prompt template, ours is still outperforms TAC with multiple prompt templates and achieves comparable performance to ours with multiple prompt templates, which implies that the performance gains do NOT come from prompt engineering. Compared with TAC, the clustering performance of ours is less sensitive to the choice of a single prompt template. Ours consistently and significantly outperforms TAC (where the kernel way is not used), which implies the advantage of our proposed NTK-based graph construction still holds with a single prompt template.

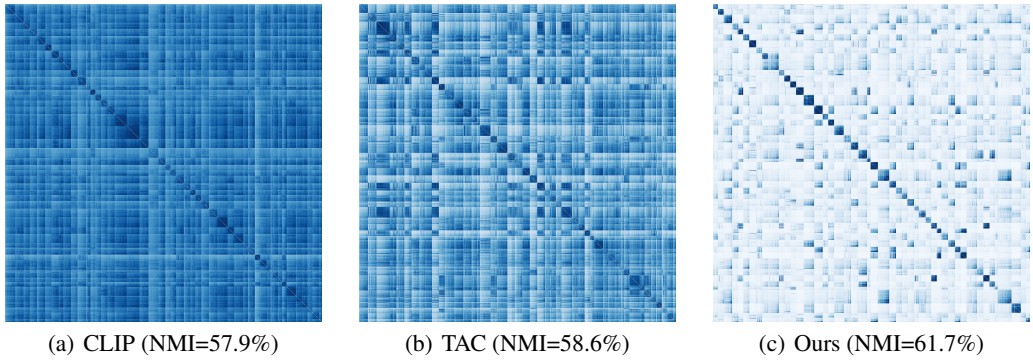

(a) CLIP (NMI=57.9%)  (b) TAC (NMI=58.6%)  (c) Ours (NMI=61.7%)

Figure 8: Visualization of affinity matrices on DTD.

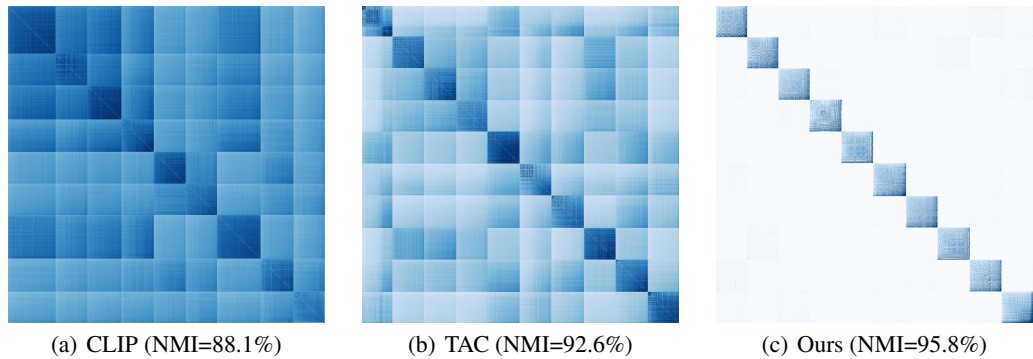

(a) CLIP (NMI=88.1%)  (b) TAC (NMI=92.6%)  (c) Ours (NMI=95.8%)

Figure 9: Visualization of affinity matrices on STL-10.

Table 8: Ablation Study on Prompt Templates

| Dataset | STL-10 | | | CIFAR-10 | | | CIFAR-20 | | | ImageNet-10 | | | ImageNet-Dogs | | | DTD | | | UCF101 | | | ImageNet-1K | | | Average | | |
|---|---|---|---|---|---|---|---|---|---|---|---|---|---|---|---|---|---|---|---|---|---|---|---|---|---|---|---|
| Metrics | NMI | ACC | ARI | NMI | ACC | ARI | NMI | ACC | ARI | NMI | ACC | ARI | NMI | ACC | ARI | NMI | ACC | ARI | NMI | ACC | ARI | NMI | ACC | ARI | NMI | ACC | ARI |
| *itap of a {}* | | | | | | | | | | | | | | | | | | | | | | | | | | | |
| TAC (Kmeans) | 91.0 | 93.8 | 87.3 | 76.9 | 80.7 | 70.5 | 56.8 | 48.7 | 36.3 | 97.0 | 97.6 | 95.8 | 70.5 | 70.5 | 57.3 | 60.2 | 45.5 | 28.4 | 80.7 | 59.8 | 50.8 | 75.1 | 46.4 | 33.6 | 76.0 | 67.9 | 57.5 |
| TAC (SC) | 93.3 | 95.9 | 93.2 | 78.5 | 88.4 | 76.5 | 53.0 | 47.2 | 35.4 | 95.1 | 97.0 | 95.8 | 75.3 | 74.6 | 61.7 | 57.5 | 42.5 | 24.1 | 81.2 | 61.4 | 54.4 | 77.4 | 48.0 | 35.6 | 76.4 | 69.4 | 59.6 |
| Ours | 94.3 | 97.5 | 94.5 | 83.2 | 90.6 | 81.1 | 62.4 | 56.3 | 39.9 | 96.9 | 98.2 | 97.4 | 82.5 | 84.5 | 71.0 | 61.6 | 51.8 | 32.7 | 81.8 | 67.5 | 58.8 | 78.6 | 54.6 | 38.9 | 80.2 | 75.1 | 64.3 |
| *a bad photo of the {}* | | | | | | | | | | | | | | | | | | | | | | | | | | | |
| TAC (Kmeans) | 88.5 | 80.2 | 78.4 | 81.8 | 90.0 | 79.3 | 60.8 | 54.6 | 41.1 | 96.0 | 97.4 | 96.4 | 73.9 | 74.5 | 62.6 | 58.2 | 44.6 | 28.5 | 80.7 | 60.0 | 50.9 | 76.9 | 47.6 | 36.0 | 77.1 | 68.6 | 59.1 |
| TAC (SC) | 92.5 | 95.0 | 90.2 | 81.3 | 90.3 | 80.1 | 54.5 | 52.7 | 38.9 | 96.0 | 97.6 | 95.4 | 64.0 | 65.6 | 52.8 | 58.5 | 43.9 | 26.8 | 81.7 | 61.9 | 55.6 | 76.8 | 48.2 | 35.7 | 75.7 | 69.4 | 59.4 |
| Ours | 95.4 | 98.1 | 95.9 | 83.1 | 91.6 | 82.1 | 62.3 | 56.4 | 40.9 | 97.1 | 98.2 | 97.2 | 82.1 | 84.3 | 70.6 | 61.5 | 52.0 | 32.0 | 82.9 | 67.5 | 58.2 | 78.9 | 55.6 | 39.4 | 80.4 | 75.5 | 64.5 |
| *a origami {}* | | | | | | | | | | | | | | | | | | | | | | | | | | | |
| TAC (Kmeans) | 90.4 | 83.3 | 82.7 | 79.6 | 87.9 | 74.8 | 58.4 | 51.4 | 37.2 | 94.8 | 95.6 | 95.1 | 73.6 | 74.3 | 62.8 | 58.3 | 44.8 | 28.4 | 80.7 | 59.9 | 50.7 | 72.2 | 38.4 | 26.8 | 76.0 | 67.0 | 57.3 |
| TAC (SC) | 93.2 | 94.4 | 91.4 | 79.1 | 88.1 | 75.4 | 55.2 | 54.1 | 39.2 | 95.6 | 97.8 | 96.9 | 72.3 | 73.5 | 61.1 | 58.1 | 44.0 | 26.9 | 81.8 | 63.2 | 55.9 | 70.2 | 40.5 | 3.5 | 75.7 | 69.4 | 56.3 |
| Ours | 94.9 | 97.8 | 95.3 | 82.9 | 91.1 | 81.7 | 63.6 | 58.7 | 40.3 | 96.7 | 98.8 | 96.7 | 81.7 | 84.5 | 70.7 | 60.6 | 51.0 | 33.0 | 83.1 | 66.6 | 57.9 | 76.5 | 53.7 | 33.1 | 80.0 | 75.3 | 63.6 |
| *a photo of the large {}* | | | | | | | | | | | | | | | | | | | | | | | | | | | |
| TAC (Kmeans) | 92.6 | 93.6 | 89.7 | 80.8 | 89.0 | 77.4 | 60.3 | 54.3 | 39.2 | 95.2 | 96.6 | 95.5 | 74.2 | 74.3 | 62.8 | 60.0 | 45.6 | 28.2 | 80.7 | 59.7 | 50.8 | 74.2 | 45.4 | 35.7 | 77.2 | 69.8 | 59.9 |
| TAC (SC) | 93.3 | 94.1 | 93.9 | 81.3 | 90.2 | 79.9 | 54.6 | 48.3 | 36.5 | 96.6 | 95.0 | 94.6 | 74.6 | 75.6 | 62.0 | 57.9 | 43.8 | 26.6 | 80.5 | 60.2 | 53.2 | 74.3 | 46.5 | 34.1 | 76.6 | 69.2 | 60.1 |
| Ours | 95.5 | 98.2 | 96.0 | 83.0 | 91.6 | 82.0 | 62.4 | 59.1 | 41.9 | 96.8 | 98.4 | 97.5 | 81.8 | 84.9 | 70.5 | 61.0 | 51.5 | 33.7 | 84.8 | 70.3 | 61.8 | 77.4 | 54.4 | 35.2 | 80.3 | 76.1 | 64.8 |
| *a {} in a video game* | | | | | | | | | | | | | | | | | | | | | | | | | | | |
| TAC (Kmeans) | 91.4 | 83.9 | 83.8 | 81.4 | 90.0 | 79.0 | 61.9 | 55.5 | 40.1 | 96.0 | 94.6 | 94.9 | 71.8 | 73.7 | 63.2 | 59.1 | 45.5 | 28.3 | 80.8 | 59.9 | 50.8 | 75.2 | 45.4 | 35.7 | 77.2 | 68.6 | 59.5 |
| TAC (SC) | 91.4 | 93.2 | 92.0 | 80.3 | 89.3 | 77.7 | 54.6 | 51.6 | 37.5 | 96.4 | 95.6 | 94.8 | 73.6 | 74.6 | 63.4 | 58.6 | 43.9 | 26.7 | 81.4 | 62.2 | 54.7 | 75.8 | 44.3 | 35.0 | 76.5 | 69.3 | 60.2 |
| Ours | 95.3 | 98.1 | 95.8 | 83.1 | 91.3 | 82.2 | 62.5 | 57.1 | 38.3 | 97.7 | 99.0 | 97.4 | 80.9 | 83.9 | 70.1 | 60.9 | 51.2 | 33.4 | 82.9 | 67.8 | 59.2 | 77.7 | 54.3 | 36.2 | 80.1 | 75.3 | 64.1 |
| *art of the {}* | | | | | | | | | | | | | | | | | | | | | | | | | | | |
| TAC (Kmeans) | 91.2 | 83.6 | 83.7 | 81.1 | 89.7 | 78.7 | 59.1 | 52.4 | 37.3 | 96.0 | 98.2 | 97.0 | 72.3 | 72.8 | 60.3 | 58.4 | 44.6 | 28.6 | 80.8 | 59.9 | 51.0 | 75.2 | 44.5 | 33.6 | 76.8 | 68.2 | 58.8 |
| TAC (SC) | 92.4 | 93.1 | 91.9 | 80.5 | 89.6 | 78.5 | 54.4 | 54.2 | 38.2 | 97.0 | 97.0 | 95.4 | 74.9 | 75.1 | 62.9 | 56.9 | 42.3 | 26.7 | 80.8 | 62.2 | 53.6 | 78.1 | 48.4 | 35.5 | 76.9 | 70.2 | 60.3 |
| Ours | 95.5 | 98.2 | 96.1 | 83.0 | 91.6 | 82.2 | 63.0 | 58.9 | 43.0 | 97.3 | 98.8 | 97.1 | 81.3 | 83.2 | 70.2 | 60.7 | 51.4 | 32.5 | 82.5 | 67.3 | 58.1 | 78.3 | 55.1 | 38.9 | 80.2 | 75.6 | 64.8 |
| *a photo of the small {}* | | | | | | | | | | | | | | | | | | | | | | | | | | | |
| TAC (Kmeans) | 87.6 | 79.9 | 77.5 | 78.0 | 86.6 | 70.9 | 60.1 | 55.2 | 39.2 | 95.9 | 96.0 | 95.0 | 74.6 | 74.1 | 63.9 | 58.0 | 44.7 | 28.3 | 80.7 | 59.9 | 50.8 | 72.2 | 38.3 | 26.6 | 75.9 | 66.1 | 56.5 |
| TAC (SC) | 92.3 | 94.8 | 90.0 | 80.7 | 89.7 | 79.0 | 55.7 | 54.3 | 40.3 | 97.0 | 97.6 | 96.4 | 74.6 | 74.0 | 63.9 | 57.3 | 43.4 | 25.2 | 80.9 | 62.0 | 54.2 | 70.1 | 40.5 | 30.5 | 76.1 | 69.6 | 59.9 |
| Ours | 94.8 | 97.8 | 95.3 | 82.9 | 90.9 | 81.4 | 63.1 | 58.8 | 41.8 | 97.6 | 98.8 | 96.3 | 82.2 | 84.0 | 70.3 | 61.7 | 51.4 | 33.3 | 83.4 | 66.9 | 57.9 | 79.1 | 56.1 | 39.3 | 80.6 | 75.6 | 64.4 |
| *Prompt Ensembling* | | | | | | | | | | | | | | | | | | | | | | | | | | | |
| TAC (Kmeans) | 92.3 | 94.5 | 89.5 | 80.8 | 90.1 | 79.8 | 60.7 | 55.8 | 42.7 | 97.5 | 98.6 | 97.0 | 75.1 | 75.1 | 63.6 | 60.1 | 45.9 | 29.0 | 81.6 | 61.3 | 52.4 | 77.8 | 48.9 | 36.4 | 78.3 | 71.3 | 61.3 |
| TAC (SC) | 92.6 | 94.3 | 94.2 | 81.2 | 90.3 | 80.1 | 56.9 | 54.5 | 30.1 | 97.0 | 98.3 | 96.8 | 75.3 | 75.8 | 64.4 | 58.6 | 44.0 | 27.1 | 79.6 | 60.0 | 50.1 | 78.0 | 49.1 | 36.2 | 77.4 | 70.8 | 59.9 |
| **Ours** | **95.8** | **98.3** | **96.3** | **83.3** | **92.0** | **83.0** | **63.3** | **59.6** | **43.5** | **97.8** | **99.2** | **98.4** | **82.4** | **84.9** | **71.4** | **61.7** | **52.0** | **33.6** | **83.0** | **67.9** | **59.4** | **79.2** | **56.3** | **39.4** | **80.8** | **76.3** | **65.6** |

# K  VALIDATION OF IMAGE DATA LEAKAGE

It should be noted that many evaluation datasets used in this work have appeared in the training of the vision-language foundation models, which can raise a data-leakage problem since the representations under interest has been well obtained. To maximally prevent data-leakage problem in the OpenAI's pretrained CLIP models, we additionally conduct experiments using the pretrained CLIP models in OpenCLIP on CIFAR-10, CIFAR-100, ImageNet-A and ImageNet-1K, where the overlap percentage with the pre-trained dataset is 0.02%, 0.03%, 0.04% and 1.02%, respectively.

Table 9: Clustering performance with OpenCLIP's pretrained models. The best results are in **bold**.

| Dataset | STL-10 | | | CIFAR-10 | | | CIFAR-20 | | | ImageNet-10 | | | ImageNet-Dogs | | | DTD | | | UCF101 | | | ImageNet-1K | | | Average | | |
|---|---|---|---|---|---|---|---|---|---|---|---|---|---|---|---|---|---|---|---|---|---|---|---|---|---|---|---|
| Metrics | NMI | ACC | ARI | NMI | ACC | ARI | NMI | ACC | ARI | NMI | ACC | ARI | NMI | ACC | ARI | NMI | ACC | ARI | NMI | ACC | ARI | NMI | ACC | ARI | NMI | ACC | ARI |
| TAC (Kmeans) | 93.1 | 95.9 | 93.4 | 84.7 | 91.9 | 83.4 | 67.4 | 63.3 | 50.3 | 97.9 | 97.0 | 96.8 | 73.4 | 73.8 | 60.3 | 65.6 | 52.3 | 36.4 | 81.1 | 61.6 | 52.9 | 76.8 | 47.5 | 34.7 | 80.0 | 72.9 | 63.5 |
| TAC (SC) | 93.6 | 96.2 | 92.8 | 86.1 | 93.3 | 85.9 | 62.0 | 57.6 | 45.3 | 97.7 | 98.0 | 96.9 | 74.4 | 74.5 | 61.2 | 65.2 | 53.8 | 36.7 | 81.2 | 64.8 | 56.2 | 76.7 | 47.3 | 35.1 | 79.7 | 73.2 | 62.8 |
| Ours | **96.1** | **99.1** | **97.3** | **89.9** | **95.4** | **90.9** | **66.5** | **64.2** | **51.8** | **97.3** | **99.6** | **98.8** | **81.5** | **83.2** | **70.7** | **66.2** | **56.4** | **40.6** | **82.8** | **67.1** | **58.0** | **78.8** | **55.0** | **39.0** | **82.4** | **77.5** | **68.4** |

Table 10: Validation of label-name leakage.

| Dataset | CIFAR-10 | | | CIFAR-20 | | | ImageNet-1K | | |
|---|---|---|---|---|---|---|---|---|---|
| Metrics | NMI | ACC | ARI | NMI | ACC | ARI | NMI | ACC | ARI |
| WordNet w/o ground-truth class name | 83.3 | 91.8 | 83.1 | 63.4 | 59.8 | 43.7 | 79.3 | 56.2 | 39.6 |
| Full WordNet | 83.3 | 92.0 | 83.0 | 63.3 | 59.6 | 43.5 | 78.8 | 56.3 | 39.4 |

It can be found from Table 9 that using OpenCLIP's checkpoints contribut to better or comparable performance than OpenAI's checkpoints, which means that performance benefit of CLIP-based clustering does not result from the data-leakage problem. Besides, we note that, using both kinds of checkpoints, our method consistently outperforms TAC.

## L  VALIDATION OF LABEL-NAME LEAKAGE

To further rule out potential label-name leakage, we additionally conducted experiment where we remove all words that exactly match any ground-truth class name from WordNet and re-run our method. As shown in Table 10, the clustering performance (averaged over 5 runs) of our method remains essentially the same (ACC / NMI / ARI differences are small), indicating that our emprical advantages do not rely on the presence of true label names but on the richer semantic structure provided by generic "in-the-wild" nouns and their interactions via our proposed NTK-based affinity.

## M  STATISTIC SIGNIFICANCE

We report the statistical significance of our method via statistical comparisons over 8 benchmarks used in Table 1 and Table 2. To this end, a very common practice yields the paired t-test . So before we list the results, let's define the following hypothesis to test.

- $p_0$ : The compared two models may not have significant statistical differences regarding their clustering performance.
- $p_1$ : The compared two models may have significant statistical differences regarding their clustering performance.

Table 11 reports the $p$-value of Wilcoxon Signed-Ranks Test where the clustering performance is measured by NMI. According to the p-values of our method against all the compared CLIP-based baselines, one can conclude our method consistently rejects the null hypothesis with $p \ll 0.05$, which means our clustering performance is statistically significant enough to be distinguished from the others.

## N  TIME AND SPACE COMPLEXITY

The time and space complexity of the proposed method can be broken down into two main components: (a) NTK-induced graph construction and (b) Regularized Affinity Diffusion (RAD).

Table 11: Paired hypothesis test $p$-values.

| | TAC | SIC | CLIP (Kmeans) |
|---|---|---|---|
| Ours | 0.0078 | 0.0039 | 0.0078 |

Table 12: Computation cost of our method and spectral clustering on ImageNet-1K.

| NTK-induced graph construction | RAD | Spectral clustering |
|---|---|---|
| 1.1 mins | 8.2 mins | 3.6 mins |

For NTK-induced graph construction, one can easily check from Eq. (8) that time and space complexity is $O(BM^2(d+N))$ and $O((M+BN)d)$ where $B$ is the number of prompt template, $d$ is the CLIP feature dimension, $N$ and $M$ is the number of positve nouns and images.

For RAD, it seems that we need space complexity is $O(BM^2)$ space to store the $B$ prompt-specific affinity matrices. However, since the affinity matrices share a sparse mutual $q$-nn pattern (each row has $O(q)$ nonzeros), the space complexity accordingly drops to $O(BMq)$, which is linearly dominated by $M$ since $q, B \ll M$. In practice, by storing the affinity matrices as Torch sparse tensor, the total memory usage is around 30 MB.

Regarding to the time complexity of RAD, it has two parts. The first one is updating $\hat{A}$ via Eq. (13). Since each $\mathbf{S}^{(b)}$ (the row-normalized $\mathbf{A}_{\text{NTK}}^{(b)}$) has $O(q)$ nonzeros, each multiplication is around $O(Mq^2)$, giving $O(t_1 BMq^2)$ for computing Eq. (13) for $t_1$ steps. The second part is updating $\beta$ in $O(t_2 B^2)$ with $t_2$-step coordinate descent shown in Appendix E. Considering the overall iteration number T in alternating optimization, the final space complexity of RAD is $O(T(t_1 BMq^2 + t_2 B^2))$. Since $t_1, t_2, T, B^2, q^2 \ll M$, we can conclude that the over time complexity is dominated by $O(M)$.

Table 12 reports computation cost of our method and spectral clustering on ImageNet-1K on a single Nvidia A100, where one can find that RAD takes 8 mins and spectral clustering takes 3.6 mins. Therefore, we argue that our method and spectral clustering can be scaled to large datasets.

