# OpenReview forum: "Delving into Spectral Clustering with Vision-Language Representations"
_ICLR.cc/2026/Conference — ICLR 2026 Poster_

### Official Review · Reviewer_kiq4 · 2025-10-21

**Soundness:** 3
**Presentation:** 3
**Contribution:** 2
**Rating:** 6
**Confidence:** 5

**Summary:**

This paper proposes a spectral clustering method for features extracted from vision language models. The main contributions lie in integrating the affinities from the vision and text modalities, as well as multiple prompts. Experimental results on 13 datasets demonstrate the effectiveness of the proposed method.

**Strengths:**

1. The proposed method is mathematically grounded, with detailed derivations provided in the appendix.
2. The paper is clearly organized and written in general.
3. Experiments show that the proposed method outperforms existing baselines on both classic and more challenging, fine-grained datasets.
4. Parameter analyses are conducted to investigate the robustness of the method.

**Weaknesses:**

1. To improve the readability of the paper, I encourage the authors to provide a figure to illustrate the proposed method, instead of purely presenting mathematical notations and equations.
2. What is the clustering performance of the NTK-induced affinity matrix $A_{NTK}$?
3. The mentioned baseline TAC has a variation that trains a clustering head to further improve the performance. Currently, it seems that the authors only compare its training-free version. A more thorough comparison is expected.
4. Ablation studies on each component of the method could be attached.
5. What is the time and space complexity of the proposed method? Considering the computational efficiency of spectral clustering methods, can the proposed method scale to large datasets? How much time does it take to perform clustering on ImageNet-1K?

**Questions:**

I expect the authors to address my concerns on the weakness section.

---

> ### Author Response · Authors · 2025-11-20
>
> We thank Reviewer kiq4 for your thorough comments. As to the weaknesses and minor issues you pointed out, we took them very seriously, and have updated parts of the paper to improve it. Our response is as follows.
>
> # 1. Provide a figure to illustrate the proposed method
>
> We agree that a visual illustration of the proposed method would significantly improve readability.  We believe the overview figure newly added in Page 3 of the revised manuscript makes the intuition behind the mathematical formulation much clearer for readers.
>
> # 2. What is the clustering performance of the NTK-induced affinity matrix $\mathbf{A}\_{\text{NTK}}$
>
> Since the NTK-induced affinity matrix $\mathbf{A}\_{\text{NTK}}$ is built upon a single prompt template we report the clustering performance of $\mathbf{A}\_{\text{NTK}}$ with different single prompt as follows. Note that, when single prompt is used, our proposed Regularized Affinity Diffusion in section 3.2 is automatically disabled.
>
> ## 1. **itap of a $\<label\>$**
>
> | CIFAR-10| NMI | ACC | ARI |
> |-|-|-|-|
> | TAC |76.9|80.7|70.5
> |  Ours |83.2|90.6|81.1
> | TAC (full model)|    80.8      |    90.1        |   79.8
> |  Ours (full model)  | 83.3           |     92.0       |   83.0
>
> | CIFAR-20| NMI | ACC | ARI |
> |-|-|-|-|
> | TAC |56.8|48.7|36.3
> |  Ours   |62.4|56.3|39.9
> | TAC (full model) |60.7|55.8|42.7
> | Ours (full model)   |63.3|59.6|43.5
>
> | ImageNet-1K| NMI | ACC | ARI |
> |-|-|-|-|
> | TAC |   75.1|46.4|33.6
> |  Ours |78.6|54.6|38.9
> | TAC (full model) |77.8|48.9|36.4|
> |  Ours (full model)  |79.2|56.3|39.4
>
> ## 2. **a bad photo of the $\<label\>$**
>
> | CIFAR-10| NMI | ACC | ARI |
> |-|-|-|-|
> |TAC |81.8|90.0|70.5
> |Ours |83.1|91.6|82.1
> |TAC (full model)|    80.8      |    90.1        |   79.8
> |Ours (full model)  | 83.3           |     92.0       |   83.0
>
> |CIFAR-20| NMI | ACC | ARI |
> |-|-|-|-|
> |TAC|60.8|54.6|41.1
> |Ours|62.3|56.4|40.9
> |TAC (full model) |60.7|55.8|42.7
> | Ours (full model)|63.3|59.6|43.5
>
> |ImageNet-1K| NMI | ACC | ARI |
> |-|-|-|-|
> |TAC|75.9|47.6|36.0
> |Ours|78.9|55.6|39.4
> |TAC (full model)|77.8|48.9|36.4|
> |Ours (full model)|79.2|56.3|39.4
>
> ## **3. a origami $\<label\>$**
>
> | CIFAR-10| NMI | ACC | ARI |
> |-|-|-|-|
> |TAC| 79.6|87.9|74.8
> | Ours |82.6|91.1|81.7
> |TAC (full model)| 80.8|90.1|79.8
> | Ours (full model) |83.3  |92.0| 83.0
>
> | CIFAR-20| NMI | ACC | ARI |
> |-|-|-|--|
> |TAC| 58.4|51.3|37.2
> |Ours|63.6|58.7|40.3
> |TAC (full model) |60.7|55.8|42.7
> |Ours (full model)|63.3|59.6|43.5
>
> | ImageNet-1K| NMI | ACC | ARI |
> |-|-|-|-|
> |TAC|72.2|38.4|16.8
> |Ours|76.5|53.7|33.1
> |TAC (full model) |77.8|48.9|36.4|
> |Ours (full model)|79.2|56.3|39.4
>
> Several observations can be found in the tables above:
> - Even with a single prompt template, ours is still outperforms TAC (full model) and achieves comparable performance with Ours (full model), which implies that the performance gains do **NOT** come from prompt engineering.
> - Compared with TAC, the clustering performance of ours is **less sensitive** to the choice of a single prompt template.
> - Ours consistently and significantly outperforms TAC (where the kernel way is not used), which implies the advantage of our proposed NTK-based graph construction still holds with a single prompt template.
>
> More experiment results regarding templates and datasets can be found in Appendix J of the revised manuscript
>
> # 3. Comparison with training-based TAC
>
> Tables below compare our method with training-based TAC, where our method achieves comparable or better clustering performance than TAC **without introducing additional trainable parameters**. We have included this comparison in Table 1 and Table 2 of the revised manuscript.
>
> | STL-10| NMI | ACC | ARI |
> |---------|--------------|----------|----------|
> | TAC (training-based)|95.5|98.2|96.1
> | Ours       | 95.8|98.3|96.3
>
> | CIFAR-10| NMI | ACC | ARI |
> |---------|--------------|----------|----------|
> | TAC (training-based)|83.3|91.9|83.1
> | Ours       | 83.3           |     92.0       |   83.0
>
> | CIFAR-20| NMI | ACC | ARI |
> |---------|--------------|----------|----------|
> | TAC (training-based)|61.1|60.7|44.8
> |  Ours              |63.3|59.6|43.5
>
> | ImageNet-10| NMI | ACC | ARI |
> |---------|--------------|----------|----------|
> | TAC (training-based)|98.5|99.5|98.3
> |  Ours        |97.8|99.2|98.4
>
> | ImageNet-Dogs| NMI | ACC | ARI |
> |---------|--------------|----------|----------|
> | TAC (training-based)|80.6|83.0|72.2
> |  Ours        |82.4|84.9|71.4

---

> ### Author Response · Authors · 2025-11-20
>
> # 4. Ablation studies on each component
>
> ## 4.1. Regularized Affinity Diffusion (RAD)
>
> We kindly note that ablation study on RAD has been conducted in Appendix F, where we consider the following two alternatives to RAD:
>
> (a) The first alternative is to naively average the obtained $B$ affinity matrices $\mathbf{A}\_{\text{NTK}}^{(1)},\ldots,\mathbf{A}\_{\text{NTK}}^{(B)}$, i.e.,
> $\hat{\mathbf{A}}=\frac{1}{B}\sum_{b=1}^B\mathbf{A}^{(b)}_{\text{NTK}}.$
>
> (b) The second alternative uses Eq. (6) and Eq. (7) to construct the ensembled affinity matrix $\hat{\mathbf{A}}$ except that $\boldsymbol{\theta}\_0=vec\left(\bar{\mathbf{W}}\right)=vec\left(\frac{1}{B}\sum\_{b=1}^B\mathbf{W}^{(b)}\right)$ where, following Prompt Ensembling used in TAC, $\mathbf{W}^{(b)}$ is the CLIP features of positive nouns induced by $b$-th prompt template $\Delta^{(b)}(\cdot)$. Formally, $\hat{\mathbf{A}}$ computed by the second alternative is given by:
>
> $$
> \hat{\mathbf{A}}[i,j]= \\begin{cases}
> \mathcal{K}\_{\bar{\mathbf{W}}}\left(f\_{\mathcal{X}}(\mathbf{x}\_i),f\_{\mathcal{X}}(\mathbf{x}\_j)\right),  & \text{ if } f\_{\mathcal{X}}(\mathbf{x}\_i)\in\mathcal{N}\_{q}(f\_{\mathcal{X}}(\mathbf{x}\_j),\mathcal{K}\_{\bar{\mathbf{W}}})\wedge f\_{\mathcal{X}}(\mathbf{x}\_j)\in\mathcal{N}\_{q}(f\_{\mathcal{X}}(\mathbf{x}\_i),\mathcal{K}\_{\bar{\mathbf{W}}}). \\\\
> 0, &\text{  otherwise. }
> \\end{cases}
> $$
>
> The tables below show that Ours (RAD) achieves the best clustering performance, which validates the effectiveness. Besides, one can also find that Ours (b) significantly outperforms TAC, which implies the effectiveness of our proposed NTK-induced affinity measure.
>
> | CIFAR-10| NMI | ACC | ARI |
> |---------|--------------|----------|----------|
> | TAC |    80.8      |    90.1        |   79.8
> |  Ours (a)  |71.4|74.7|56.8
> | Ours (b) |82.9| 91.3| 82.4
> |  Ours (full model)       | 83.3           |     92.0       |   83.0
>
> | CIFAR-20| NMI | ACC | ARI |
> |---------|--------------|----------|----------|
> | TAC |60.7|55.8|42.7
> |  Ours (a)  |45.1|42.6|29.9
> | Ours (b) |60.9|55.0|43.4
> |  Ours (full model)       |63.3|59.6|43.5
>
> | ImageNet-1K| NMI | ACC | ARI |
> |---------|--------------|----------|----------|
> | TAC |77.8|48.9|36.4
> |  Ours (a)  |77.1|51.0|35.4
> | Ours (b) |79.2|53.3|38.4
> |  Ours (full model) |79.2|56.3|39.4
>
> ## 4.1. Prompt templates
>
> Please refer to our second response for the ablation study in prompt template.
>
> # 5. Can the proposed method scale to large datasets
>
> The time and space complexity of the proposed method can be broken down into two main components: (a) NTK-induced graph construction and (b) ​Regularized Affinity Diffusion (RAD)​.
>
> For NTK-induced graph construction, one can easily check from Eq. (8) that time and space complexity is $O(BM^2(d+N))$ and $O((M+BN)d)$ where $B$ is the number of prompt template, $d$ is the CLIP feature dimension, $N$ and $M$ is the number of positve nouns and images.
>
> For RAD, it seems that we need space complexity is $O(BM^2)$ space to store the $B$ prompt-specific affinity matrices. However, since the affinity matrices share a sparse mutual $q$-nn pattern (each row has $O(q)$ nonzeros) and can be implemented as **Torch sparse tensors**, the space complexity accordingly **drops to $O(BMq)$**, which is **linearly dominated** by $M$ since $q,B\ll M$.
>
> Regarding to the time complexity of RAD, it has two parts. The first one is updating $\hat{A}$via Eq. (13). Since each $\mathbf{S}^{(b)}$ (the row-normalized $\mathbf{A}^{(b)}_{\text{NTK}}$) has $O(q)$ nonzeros, each multiplication is around $O(Mq^2)$, giving $O(t_1BMq^2)$ for computing Eq. (13) for $t_1$ steps . The second part is updating $\beta$ in $O(t_2B^2)$ with $t_2$-step coordinate descent shown in Appendix E. Considering the overall iteration number T in alternating optimization, the final space complexity of RAD is $O(T(t_1BMq^2+t_2B^2))$.  Since
> $t_1, t_2, T, B^2, q^2 ≪ M$, we can conclude that the over time complexity is dominated by $O(M)$.
>
> The table below reports computation cost of our method on ImageNet-1K on a single Nvidia A100, where one can find that RAD takes 8 mins and spectral clustering takes 3.6 mins. Therefore, we argue that our method and spectral clustering can be scaled to large datasets.
>
> | NTK-induced graph construction| RAD| spectral clustering|
> |---------|--------------|----------|
> |1.1min|8.2min|3.6min|

---

> > ### Author Response · Authors · 2025-11-25
> >
> > Dear Reviewer,
> >
> > The rebuttal phase has opened, and we have submitted a detailed response to your comments. Thank you again for your valuable feedback and for the time you have devoted to our paper.
> >
> > When convenient, **could you please take a look at our rebuttal and let us know if our responses adequately address your concerns? If you feel that the issues have been resolved, we would be sincerely grateful if you could consider increasing your score accordingly.**
> >
> > Your feedback would be extremely helpful for us to further refine the work.
> >
> > Thank you again for your time and thoughtful review.
> >
> > Kind regards,
> >
> > Senior Author

---

> > ### Comment · Reviewer_kiq4 · 2025-11-27
> >
> > Thanks for the responses. My concerns have been addressed.

---

> > > ### Author Response · Authors · 2025-11-27
> > >
> > > Thank you very much for your response and for confirming that all concerns have been addressed. If it is possible to **further increase the score** based on the updated clarifications and results, we would sincerely appreciate it. Thank you again for your time and consideration.

---

### Official Review · Reviewer_rFTU · 2025-10-31

**Soundness:** 4
**Presentation:** 4
**Contribution:** 3
**Rating:** 8
**Confidence:** 3

**Summary:**

This paper is about a new clustering approach named Neural Tangent Kernel Spectral Clustering. This approach that extends traditional spectral clustering to leverage multi modal repreentations from pre trained vision language models. The core contribution is formulating image affinity as a coupling of visual proximity and semantic overlap through anchoring neural tangent kernels with semantically relevant positive nouns. The paper includes a regularized affinity diffusion mechanism to ensemble affinity matrices from different prompts and demonstrate substantial improvements across 16 datasets.

**Strengths:**

+ The integration of neural tangent kernel theory with vision language representations for spectral clustering is novel. The idea of anchoring NTK with positive nouns to create semantically aware affinity matrices represents a way to incorporate linguistic priors into clustering.
+ The paper includes a comprehensive experimental validation, where testing is performed on 16 benchmarks. The consistent improvements suggest that the method is generalizable.
+ The presentation is excellent, the paper is easy to read and well written.

**Weaknesses:**

- The method uses some positive nouns that are semantically close to images of interest. This could limit practical applicability and introduce some bias. The paper should provide some analysis about it.
- A minor weakness, is that the main paper is condensed and the related works in that are limited.

**Questions:**

- How does performance degrade with noisy or out of distribution images?

---

> ### Author Response · Authors · 2025-11-20
>
> We thank reviewer rFTU for your valuable comments. We updated the submission accordingly. Please kindly find the detailed responses below.
>
> # 1. The method uses some positive nouns that are semantically close to images of interest. This could limit practical applicability and introduce some bias
>
> We thank the reviewer for raising this important point. We agree that using positive nouns that are semantically close to the images of interest may introduce data bias. However, this limitation **​is inherent to all CLIP-based clustering methods (e.g., TAC and SIC) that rely on positive nouns**​, because in the fully unsupervised setting ground-truth class names are not accessible and one has to resort to pseudo label surrogates constructed from text corpora.
>
> To ​**analyze how this potential bias affects clustering**​, we conduct a domain-generalization study in Sec. 4.4. Specifically, we select positive nouns based only on the train split of ImageNet-1K and then evaluate clustering performance on several ImageNet-1K variants with distribution shifts: ImageNet-C (**noisy** images), ImageNet-V2 (**new test distribution**), and ImageNet-S (**out-of-distribution** images). For your convenience, we provide the results in the below.
>
> | Ours| NMI | ACC | ARI |
> |---------|--------------|----------|----------|
> | ImageNet-1K|79.2|56.3|39.4
> | ImageNet-C  | 75.0|44.0|28.1
> |  ImageNet-V2  |74.5|42.7|27.2
> | ImageNet-S|72.7|40.1|25.3
>
> | TAC| NMI | ACC | ARI |
> |---------|--------------|----------|----------|
> | ImageNet-1K|77.8|48.9|36.4
> | ImageNet-C  |71.4|39.2|25.6
> |  ImageNet-V2  |71.7|38.5|23.0
> | ImageNet-S|70.7|34.8|22.1
>
> The results show that both TAC and our method experience performance drops under these shifts, confirming that data bias can indeed impact CLIP-based clustering. Even so, our method consistently outperforms TAC on all three datasets, indicating **stronger robustness** to such bias and domain shift.
>
> # 2. The main paper is condensed and the related works in that are limited
>
> We thank the reviewer for pointing this out. The main paper is indeed quite condensed because we prioritized presenting the core methodology, theoretical analysis, and extensive experiments within the strict page limit.
>
> As noted in the original submission, a more detailed discussion of related work—including deep spectral clustering, affinity diffusion, and CLIP-based vision–language clustering—was provided in Appendix A.
>
> Since the page limit will be increased to 10 pages during the discussion/rebuttal phase, we have moved the content currently in Appendix A into the main body of the revised manuscript.

---

> > ### Author Response · Authors · 2025-11-25
> >
> > Dear Reviewer,
> >
> > The rebuttal phase has opened, and we have submitted a detailed response to your comments. Thank you again for your valuable feedback and for the time you have devoted to our paper.
> >
> > When convenient, **could you please take a look at our rebuttal and let us know if our responses adequately address your concerns? If you feel that the issues have been resolved, we would be sincerely grateful if you could consider increasing your score accordingly.**
> >
> > Your feedback would be extremely helpful for us to further refine the work.
> >
> > Thank you again for your time and thoughtful review.
> >
> > Kind regards,
> >
> > Senior Author

---

### Official Review · Reviewer_nmSD · 2025-10-31

**Soundness:** 2
**Presentation:** 3
**Contribution:** 2
**Rating:** 4
**Confidence:** 4

**Summary:**

The paper proposes Neural Tangent Kernel Spectral Clustering (NTK-SC) for image clustering with vision–language models.
The core idea is to anchor a proxy network at text features of positive nouns, compute an empirical NTK over CLIP image embeddings, and use it as an affinity for spectral clustering.

Concretely, with CLIP's unit-normalized visual features \\(z=f_X(x)\in\mathbb{R}^d\\) and noun features \\(W=[w_1,\dots,w_N]\\), the proxy is chosen as a log-sum-exp \\(g_{\theta_0}(z)=\log\sum_{k=1}^N e^{w_k^\top z/\tau}\\) (Eq. (7)). The empirical NTK between two images has the closed-form (Eq. (8)).

Affinities are sparsified via mutual \\(q\\)-NN and fed to SC (Eq. (2)). RAD alternates a diffusion update

$$
\hat A \leftarrow \sum_{b=1}^B \frac{\beta[b]}{\mu+1}\,S^{(b)}\hat A S^{(b)\top}+\frac{\mu}{\mu+1}\,E
$$

(Eqs. (11)-(13)) and a Lasso-like weight update for \\(\beta\\) (Eq. (14)), with \\(E=I\\). Experiments on 16 datasets claim consistent gains over TAC/SIC/CLIP baselines.

**Strengths:**

- The paper is easy to follow.
- Eq. (8) decomposes the affinity into visual proximity and a text-induced overlap. It is easy to compute once noun logits are available.
- The RAD update and its fixed-point interpretation are derived and come with a convergence argument for the linearized step.
- Results cover classic, fine-grained, and domain-shift benchmarks. Ablations on \\(\tau,q,\mu,\lambda\\) are provided.

**Weaknesses:**

> 1. I wonder if negative weights violate standard SC assumptions.

Since CLIP features lie on the unit sphere, \\(\langle z_i,z_j\rangle\in[-1,1]\\) (Eq. (1)). With Eq. (8), the NTK inherits negative values whenever the cosine is negative:

$$
K_{\theta_0}(z_i,z_j)=\frac{1}{\tau^2}\langle z_i,z_j\rangle \sum_k s_i[k]s_j[k].
$$

Yet the method sets \\(A_{ij}=K_{\theta_0}\\) on mutual \\(q\\)-NN (Eq. (6)) without rectification or signed-graph treatment. Normalized-cut SC (Eq. (2)) typically presumes nonnegative affinities, negative edges break stochastic interpretations and many guarantees. No analysis or ablation is given for rectifying \\(A\ge 0\\) (e.g., \\(A\leftarrow\max(0,K)\\) or shifting).

> 2. Graph quality and connectivity under fixed q are unexamined.

The paper fixes \\(\tau=0.04,\,q=30,\,\mu=0.1,\,\lambda=10\\) for all datasets (sizes and densities vary widely). There is no report of connected components or sensitivity on large graphs (e.g., ImageNet-1K). Disconnected graphs produce multiple zero eigenvalues and can distort clustering.

> 3.Scaling and runtime are not discussed.

The RAD fixed-point (Eq. (12)) is acknowledged infeasible to invert (\\(M^2\times M^2\\)), hence an iterative update (Eq. (13)). However, per-iteration costs with \\(B\\) prompt graphs and sparse matrices (expected \\(O(B\cdot M q^2)\\)) and memory footprints are not reported

> 4.Comparisons omit crucial baselines and uncertainty.

Absent is a fair product-kernel baseline (cosine \\(\times\\) text overlap).

> 5. Evidence issues

The paper claims consistent SOTA improvements and sharper block diagonals, but (i) the evidence rests on fixed hyperparameters across datasets of very different statistics, (ii) TAC multi-prompt usage is argued via an implementation detail (footnote) rather than a principled comparison, and (iii) no statistical significance is provided.

**Questions:**

In addition to the issues raised above, I have the following questions:

1. Do you ever rectify \\(K_{\theta_0}\\) or clip \\(A\\) to be nonnegative before computing \\(L=I-D^{-1/2}AD^{-1/2}\\)? Do you remove the diagonal prior to eigen-decomposition? Please report results with \\(A\leftarrow\max(0,K_{\theta_0})\\), \\(A\leftarrow (K_{\theta_0}+|K_{\theta_0}|)/2\\), and with/without diagonal. (Eqs. (2), (6), (8))

2. What exactly is the algorithm for selecting “positive nouns,” especially given the definition “similar to any ID label”? What signals from the train split are used to filter nouns, and how do you prevent label-name leakage?

---

> ### Author Response · Authors · 2025-11-20
>
> We appreciate the constructive comments provided by Reviewer nmSD. As to the questions and suggestions you raise, we took them seriously. Our response is as follows.
>
> # 1. The paper fixes same hyper-parameters for different datasets.
>
> While we agree that it is technically feasible to tune hyper-parameters per dataset, it is significantly more reasonable to fix the same hyperparameters across datasets. This is because using shared hyperparameters also **simplifies reproducibility** and **ensures that performance differences are due primarily to the method and dataset, not to per-dataset tuning effort**.
>
> The sensitivity study in Figures 3,4,5,6 indicates that our method is robust to these choices, which is why a single global setting works well on 16 datasets. This is also reflected in the consistently strong results in Tables 1–5, where the same hyper-parameters are used everywhere. The above empirical evidence implies that our method is a **hyperparameter-light** method.
>
> # 2. There is no report of sensitivity on large graphs (e.g., ImageNet-1K).
>
> We believe the term "sensitivity" mentioned by the reviewer refers to hyper-parameter sensitivity.
>
> On this basis, we explore hyper-parameter sensitivity of our method on ImageNet-1K and then report the results in the following tables, where the robustness of our method to hyper-parameters still holds on ImageNet-1K.
>
> | $\lambda$| NMI | ACC | ARI |
> |-|-|-|-|
> | 5|78.1|54.3|37.2
> | 10|79.2|56.3|39.4
> | 15|78.9|55.8|39.0
> | 20|78.5|55.1|38.1
> | 30|78.0|54.4|37.4
> | 60|77.8|53.7|36.9
>
> | $\mu$| NMI | ACC | ARI |
> |-|-|-|-|
> | 0.1|79.2|56.3|39.4
> | 0.2|78.8|55.7|39.0
> | 0.3|78.4|55.2|38.8
> | 0.5|78.0|54.8|38.1
> | 0.7|77.6|54.3|37.6
> | 0.9|77.2|53.9|37.1
>
> | $\tau$| NMI | ACC | ARI |
> |-|-|-|-|
> | 0.02|78.0|55.1|38.5
> | 0.03|78.5|55.8|39.0
> | 0.04|79.2|56.3|39.4
> | 0.06|79.6|57.0|39.8
> | 0.08|79.0|56.0|39.1
> | 0.1|78.4|55.7|38.6
>
> | $q$| NMI | ACC | ARI |
> |-|-|-|-|
> | 8|78.4|54.7|38.0
> | 12|78.7|55.4|38.6
> | 16|78.9|55.9|38.9
> | 20|79.1|56.1|39.1
> | 30|79.2|56.3|39.4
> | 40|79.1|55.9|39.1
>
> # 3. Scaling and runtime are not discussed.
>
> As for space complexity, it seems that Eq. (13) needs $O(BM^2)$ space to store $\mathbf{S}^{(b)} (b=1,...,B)$. However, since each $\mathbf{S}^{(b)}$ has a sparse mutual $q$-nn pattern (each row has $O(q)$ nonzeros), the space complexity accordingly **drops to $O(BMq)$**, which is **linearly dominated** by $M$ since $q,B\ll M$. In practice, by storing each $\mathbf{S}^{(b)}$ as a **Torch sparse tensor**, the **total memory usage** is around 30MB.
>
> We agree with the reviewer that the time complexity of Eq. (13) is $O(BMq^2)$. In addition, we would like to note that, since $q^2\ll M$, the time complexity of Eq. (13) is linearly dominated by $M$. Thanks to our torch-based implementation where each $\mathbf{S}^{(b)}$ is stored as a **Torch sparse tensor**, the computation of Eq. (13) can be significantly speeded up by GPUs to **only takes approximately 30s on a Nvidia A100**.
>
> # 4. What exactly is the algorithm for selecting “positive nouns”
>
> ### Step 1: Over-clustering on Images
>
> - Perform **over-clustering** on image embeddings to obtain $L$ clusters
> - Each cluster center is denoted as $s_l$ where $l = 1, ..., L$
>
> ### Step 2: Reverse Probability Calculation
>
> - For each WordNet noun $t_i$, compute the probability of belonging to each semantic center $s_l$:
>
> $$
> p(y=l|\mathbf{t\_i}) = \frac{\exp(\text{sim}(t\_i, s\_l))}{\sum\_{j=1}^L \exp(\text{sim}(t\_i, s\_j))}
> $$
>
> ### Step 3: Top Noun Selection
>
> - For each cluster center $s_l$:
>   - Select nouns with **top-$\beta$ probability** of belonging to $s_l$
>   - $\beta$ controls the number of most discriminative nouns per cluster
>
> ### Step 4: Positive Noun Set Formation
>
> - Combine all selected nouns from all clusters to form the final set of positive nouns

---

> ### Author Response · Authors · 2025-11-20
>
> # 5. No statistical significance is provided
>
> As per your advice, we report the statistical significance of our method via Statistical Comparisons over 8 Data Sets used in Tables 1 and 2. To this end, a very common practice yields the paired t-test [1]. So before we list the results, let's define the following hypothesis to test.
>
> - $p_0$ : The compared two models may **not** have significant statistical differences regarding their clustering performance.
> - $p_1$ : The compared two models may have significant statistical differences regarding their clustering performance.
>
> The table below reports the $p$-value of Wilcoxon Signed-Ranks Test where the clustering performance is measured by NMI.
>
> |$p$-value| TAC | SIC| CLIP (kmeans)|
> |---------|--------------|----------|----------|
> | Ours|0.0078|0.0039|0.0078
>
> According to the p-values of our method against all the compared CLIP-based baselines, one can conclude our method consistently rejects the **null hypothesis** with $p\ll0.05$, which means our clustering performance is statistically significant enough to be distinguished from the others.
>
> [1]  Statistical Comparisons of Classifiers over Multiple Data Sets, JMLR
>
> ## Being not reported by recent articles
>
> It is notable that, to the best of our knowledge, **none** of the related articles on recent top-tiered venues tests and reports statistical significance. This is basically because a machine learning paper usually has limited testing datasets. Directly reading the actual results would be sufficient to come up with a decision if a model outperforms the SoTA.
>
> Although we have already proved our statistical significance and **are willing to include the results in the revised manuscript**, we insist that **our original experiments are self-contained and comprehensive** since we have already reported all metrics and benchmarks that have been used in the recent clustering papers that have gone through peer review.
>
> On this basis, we believe results on statistical significance are **just a plus**. As this paper focuses on a novel model proposal, discussing the potential problems of the experimental convention agreed by all other papers would be a bit out-of-scope. It could be better to have a standalone paper discussing this issue in the future.
>
> # 6. Label-name leakage
>
> We appreciate the concern and clarify that our method does not use the dataset’s class names at any stage. The pool of candidate nouns is a fixed, dataset-agnostic vocabulary collected from WordNet as in TAC and SIC. Positive nouns are selected automatically based solely on their CLIP similarity to unlabeled training images, without any access to label strings or label–image pairs.
>
> The effectiveness of our method across ​**16 diverse benchmarks**​ further supports the claim that we are not simply 'cheating' by leaking label names. The same general WordNet source and selection process works universally because it taps into general visual semantics, not dataset-specific labels.
>
> To further rule out potential label-name leakage, we additionally conducted experiment where we **remove all words that exactly match any ground-truth class name** from WordNet and re-run our method. The performance (averaged over 5 runs) remains essentially the same (ACC / NMI / ARI differences are small), indicating that our emprical advantages do not rely on the presence of true label names but on the richer semantic structure provided by generic “in-the-wild” nouns and their interactions via our proposed NTK-based affinity.
>
> | CIFAR-10| NMI | ACC | ARI |
> |---------|--------------|----------|----------|
> | WordNet w/o ground-truth class name|83.3|91.8|83.1
> |  Full WordNet|83.3|92.0|83.0
>
> | CIFAR-20| NMI | ACC | ARI |
> |---------|--------------|----------|----------|
> | WordNet w/o ground-truth class name|63.4|59.8|43.7
> |  Full WordNet|63.3|59.6|43.5
>
> | ImageNet-1K| NMI | ACC | ARI |
> |---------|--------------|----------|----------|
> | WordNet w/o ground-truth class name|79.3|56.2|39.6
> |  Full WordNet|79.2|56.3|39.4

---

> ### Author Response · Authors · 2025-11-20
>
> # 7. TAC multi-prompt usage is argued via an implementation detail (footnote) rather than a principled comparison.
>
> We apologize for the confusion. Our intention in discussing TAC’s multi-prompt usage is not to argue based on a “hidden implementation detail”, but to **ensure a fair and reproducible comparison**. While the TAC paper presents a single prompt template as the running example, the official implementation by the authors enables 7 prompt templates by default and aggregates their outputs. This is the configuration under which TAC achieves its best reported performance, and we follow it faithfully rather than introducing any additional tricks.
>
> Our experimental protocol is principled rather than relying on an accidental detail:
>
> - For both TAC and our method, we use **exactly the same set of text templates**. In this sense, the multi-prompt setting is a **shared ​evaluation budget**, **NOT** an asymmetric advantage that we only give to our method.
> - Our approach does not rely on additional textual supervision or a richer prompt pool. We operate on the **same text features** and focus on how to construct and fuse graphs more effectively (via the NTK-based affinity and RAD mechanism). In other words, multi-prompting simply provides a common set of views; our contribution lies in how these views are integrated into a better affinity structure, which is orthogonal to whether TAC is used with one or multiple prompts.
> - As we will show later, our method continues to outperform TAC in the single-prompt setup, indicating that the improvements do not stem from this implementation detail.
>
> # 8. Absent is a fair product-kernel baseline
>
> We believe the baseline the reviewer mentioned refers to the direct use of the NTK-induced affinity matrix $\mathbf{A}_{\text{NTK}}$ in for spectral clustering.
>
> ## 1. **itap of a $\<label\>$**
>
> | CIFAR-10| NMI | ACC | ARI |
> |-|-|-|-|
> | TAC |76.9|80.7|70.5
> |  Ours |83.2|90.6|81.1
> | TAC (full model)|    80.8      |    90.1        |   79.8
> |  Ours (full model)  | 83.3           |     92.0       |   83.0
>
> | CIFAR-20| NMI | ACC | ARI |
> |-|-|-|-|
> | TAC |56.8|48.7|36.3
> |  Ours   |62.4|56.3|39.9
> | TAC (full model) |60.7|55.8|42.7
> | Ours (full model)   |63.3|59.6|43.5
>
> | ImageNet-1K| NMI | ACC | ARI |
> |-|-|-|-|
> | TAC |   75.1|46.4|33.6
> |  Ours |78.6|54.6|38.9
> | TAC (full model) |77.8|48.9|36.4|
> |  Ours (full model)  |79.2|56.3|39.4
>
> ## 2. **a bad photo of the $\<label\>$**
>
> | CIFAR-10| NMI | ACC | ARI |
> |-|-|-|-|
> |TAC |81.8|90.0|70.5
> |Ours |83.1|91.6|82.1
> |TAC (full model)|    80.8      |    90.1        |   79.8
> |Ours (full model)  | 83.3           |     92.0       |   83.0
>
> |CIFAR-20| NMI | ACC | ARI |
> |-|-|-|-|
> |TAC|60.8|54.6|41.1
> |Ours|62.3|56.4|40.9
> |TAC (full model) |60.7|55.8|42.7
> | Ours (full model)|63.3|59.6|43.5
>
> |ImageNet-1K| NMI | ACC | ARI |
> |-|-|-|-|
> |TAC|75.9|47.6|36.0
> |Ours|78.9|55.6|39.4
> |TAC (full model)|77.8|48.9|36.4|
> |Ours (full model)|79.2|56.3|39.4
>
> ## **3. a origami $\<label\>$**
>
> | CIFAR-10| NMI | ACC | ARI |
> |-|-|-|-|
> |TAC| 79.6|87.9|74.8
> | Ours |82.6|91.1|81.7
> |TAC (full model)| 80.8|90.1|79.8
> | Ours (full model) |83.3  |92.0| 83.0
>
> | CIFAR-20| NMI | ACC | ARI |
> |-|-|-|--|
> |TAC| 58.4|51.3|37.2
> |Ours|63.6|58.7|40.3
> |TAC (full model) |60.7|55.8|42.7
> |Ours (full model)|63.3|59.6|43.5
>
> | ImageNet-1K| NMI | ACC | ARI |
> |-|-|-|-|
> |TAC|72.2|38.4|16.8
> |Ours|76.5|53.7|33.1
> |TAC (full model) |77.8|48.9|36.4|
> |Ours (full model)|79.2|56.3|39.4
>
> Several observations can be found in the tables above:
> - Even with single prompt template, ours is still outperforms TAC (full model) and achieves comparable performance with Ours (full model), which implies that the performance gains do **NOT** come from prompt engineering.
> - Compare with TAC, the clustering performance of ours is **less sensitive** to the choice of single prompt template.
> - Ours consistently and significantly outperforms TAC (where the kernel way is not used), which implies the advantage of our proposed NTK-based graph construction still holds with a single prompt template.
>
> More experiment results regarding templates and datasets can be found in Appendix J of the revised manuscript.

---

> ### Author Response · Authors · 2025-11-20
>
> # 9. Do you ever rectify $\mathcal{K}\_{{\theta}\_{0}}$ or clip $\mathbf {A}\_ {\text{NTK}}$
>
> We are sorry for confusion. Indeed, to avoid negativeness, our practical implementation applys min-max normalization to scale the range of $\mathcal{K}\_{{\theta}\_{0}}$ to $[0,1]$ while preserving the overall relative magnitude. We will explicitly mention this engineering trick in the revised manuscript.
>
> The table below reports clustering performace of rectifying $\mathcal{K}\_{{\theta}\_{0}}$ (which is equivalent to clipping $\mathbf {A}\_ {\text{NTK}}$ since $\mathbf {A}\_ {\text{NTK}}$ itself is a mutual $q$-nn graph and therefore symmetric.)
>
> | CIFAR-10| NMI | ACC | ARI |
> |-|-|-|-|
> | TAC |80.8|90.1|79.8
> | Ours (rectifying $\mathcal{K}\_{{\theta}\_{0}}$) |82.7|91.2|81.8
> |  Ours (min-max normalization) |83.3|92.0|83.0
>
> | CIFAR-20| NMI | ACC | ARI |
> |-|-|-|-|
> | TAC |60.7|55.8|42.7
> | Ours (rectifying $\mathcal{K}\_{{\theta}\_{0}}$) |61.2|58.9|43.0
> |  Ours (min-max normalization) |63.3|59.6|43.5
>
> | ImageNet-1K| NMI | ACC | ARI |
> |-|-|-|-|
> | TAC |77.8|48.9|36.4|
> | Ours (rectifying $\mathcal{K}\_{{\theta}\_{0}}$)|79.2|52.6|39.4
> |  Ours (min-max normalization) |79.2|56.3|39.4
>
> The tables above show that min-max normalization contributes to better.
>
> # 10. Do you remove the diagonal prior to eigen-decomposition
>
> We kindly note that this paper implements the normalized graph cut using the `SpectralClustering` function from the publicly available `scikit-learn` library, where diagonal entries of the input adjacency matrix are ignored in Laplace calculation. Therefore, there is no need to manually remove the diagonal prior before feeding into the `SpectralClustering` function.

---

> > ### Author Response · Authors · 2025-11-25
> >
> > Dear Reviewer,
> >
> > The rebuttal phase has opened, and we have submitted a detailed response to your comments. Thank you again for your valuable feedback and for the time you have devoted to our paper.
> >
> > When convenient, **could you please take a look at our rebuttal and let us know if our responses adequately address your concerns? If you feel that the issues have been resolved, we would be sincerely grateful if you could consider increasing your score accordingly.**
> >
> > Your feedback would be extremely helpful for us to further refine the work.
> >
> > Thank you again for your time and thoughtful review.
> >
> > Kind regards,
> >
> > Senior Author

---

> ### Author Response · Authors · 2025-11-26
>
> Dear Reviewer,
>
> The rebuttal phase has opened, and we have submitted a detailed response to your comments. Thank you again for your valuable feedback and for the time you have devoted to our paper.
>
> When convenient, could you please take a look at our rebuttal and let us know if our responses adequately address your concerns? If you feel that the issues have been resolved, we would be sincerely grateful if you could consider increasing your score accordingly.
>
> Your feedback would be extremely helpful for us to further refine the work.
>
> Thank you again for your time and thoughtful review.
>
> Kind regards,
>
> Senior Author

---

### Official Review · Reviewer_KbNX · 2025-11-01

**Soundness:** 2
**Presentation:** 3
**Contribution:** 2
**Rating:** 4
**Confidence:** 4

**Summary:**

This paper aims to take advantage of multi-modal (i.e., vision-language) representations of data for spectral clustering. Concretely, given a pre-trained vision-language model, data without any supervision can be anchored using neural tangent kernel with positive nouns, so that data representations can be further refined by considering the language modality of interest. This method can also take benefit from the ensemble of multiple language prompts. Thereafter, the refined affinity matrices can be used to do spectral clustering and better performance can be observed on various image datasets.

**Strengths:**

1) The problem of considering multiple modalities of data in spectral clustering is interesting and on time given the current advances in multi-modal foundation models.

2) Using neural tangent kernel to anchor the data using multi-modal information is interesting and sound.

3) Better clustering performance can be observed on various image data.

**Weaknesses:**

1) To consider both vision and language modalities in spectral clustering of images, this work extends existing techniques/strategies of "positive nouns", "neural tangent kernel", etc, which is making this work's technical contribution moderate.

2) It should be noted that many evaluation datasets used in this work have appeared in the training of the vision-language foundation models, which can raise a data-leakage problem since the representations under interest has been well obtained. Therefore, it is necessary to include a discussion on if the performance benefit is directly from the foundation model's representation or from the proposed method. Another potential way is using data not included in the training of the foundation model for clustering evaluation.

3) An ablation study on the contributions of each component in the proposed method would be interesting. For example, how much has been the ensemble contributing to the performance compared to using only one prompt? How much does a single prompt contribute using the proposed kernel way vs without using the kernel way.

4) Some typos can be fixed, e.g., "aim to grouping".

**Questions:**

Questions can be found in the above weakness section.

---

> ### Author Response · Authors · 2025-11-20
>
> We appreciate the insightful comments provided by Reviewer KbNX. Please see our responses to your concerns below.
>
> # 1. Technical contribution
>
> While we agree that our method is partially built upon prior ideas such as positive nouns and neural tangent kernels (NTK), we respectfully argue that our contribution goes beyond a straightforward combination ofexisting techniques.
>
> - **Connecting SC and NTK in a tailored way.** To the best of our knowledge, spectral clustering (SC) and NTK have evolved mostly independently in the literature. This paper constitutes the first attempt to leverage the benefits of NTK specifically for SC by proposing an SC-targeted formulation of the proxy network $g_{\boldsymbol{\theta}_{0}}(\cdot)$ with solid theoretical justification.
> - **A new way of using positive nouns for clustering.** Although prior works use positive nouns to construct an image–text-concentrated feature space, our results in Tables 1 and 2 show that this type of usage is suboptimal for SC. In contrast, our method leverages NTK from a gradient-based perspective, using the neural tangent kernel to anchor the data with multi-modal information and directly improve the affinity matrix for SC. As also noted by the reviewer in the Strengths section, this design is both interesting and sound.
>
> On this basis, and given the consistent improvements over strong CLIP-based baselines on 16 benchmarks, we believe the paper offers a meaningful and timely contribution by providing new SC-specific insights for vision–language models and demonstrating clear practical performance gains.
>
> # 2. Data-leakage Problem
>
> To maximally prevent data-leakage problem in the OpenAI's pretrained CLIP models, we additionally conduct experiments using the pretrained CLIP models in OpenCLIP [1] on CIFAR-10, CIFAR-100, ImageNet-A and ImageNet-1K, where, as shown in Table 3 of [1], the overlap percentage with the pre-trained dataset is 0.02\%, 0.03\%, 0.04\% and 1.02\%, respectively.
>
> | CIFAR-10| NMI | ACC | ARI |
> |---------|--------------|----------|----------|
> | TAC (OpenAI)|    80.8      |    90.1        |   79.8
> |  Ours (OpenAI) |     83.3           |     92.0       |   83.0
> | TAC (OpenCLIP)|84.7|91.9|83.4
> |  Ours (OpenCLIP)       |88.0|94.4|87.5
>
> | CIFAR-20| NMI | ACC | ARI |
> |---------|--------------|----------|----------|
> | TAC (OpenAI)|  60.7|55.8|42.7
> |  Ours (OpenAI) |  63.3|59.6|43.5
> | TAC (OpenCLIP)| 62.0|57.6|45.3
> |  Ours (OpenCLIP)       |66.5|64.2|51.8
>
> | ImageNet-A| NMI | ACC | ARI |
> |---------|--------------|----------|----------|
> | TAC (OpenAI)| 50.6|21.7|10.9
> |  Ours (OpenAI) |51.8|24.9|12.0
> | TAC (OpenCLIP)|50.5|22.0|10.6
> |  Ours (OpenCLIP) | 52.1|25.4|11.9
>
> | ImageNet-1K| NMI | ACC | ARI |
> |---------|--------------|----------|----------|
> | TAC (OpenAI)|78.0|48.9|36.4
> |  Ours (OpenAI) | 79.2|56.3|39.4
> | TAC (OpenCLIP)|77.8|48.5|35.7
> |  Ours (OpenCLIP) |78.8|56.0|39.0
>
> Experiments on more datasets can be found in Appendix K. It can be found that using OpenCLIP's checkpoints contribut to better or comparable performance than OpenAI's checkpoints, which means that performance benefit of CLIP-based clustering does not result from the data-leakage problem. Besides, we note that, using both kinds of checkpoints, our method consistently outperforms TAC.
>
> [1] Reproducible scaling laws for contrastive language-image learning, CVPR23, 1276 citations

---

> ### Author Response · Authors · 2025-11-20
>
> # 3. Ablation study on each component
>
> ## 3.1. Regularized Affinity Diffusion (RAD)
>
> We kindly note that ablation study on RAD has been conducted in Appendix F, where we consider the following two alternatives to RAD:
>
> (a) The first alternative is to naively average the obtained $B$ affinity matrices $\mathbf{A}\_{\text{NTK}}^{(1)},\ldots,\mathbf{A}\_{\text{NTK}}^{(B)}$, i.e.,
> $\hat{\mathbf{A}}=\frac{1}{B}\sum_{b=1}^B\mathbf{A}^{(b)}_{\text{NTK}}.$
>
> (b) The second alternative uses Eq. (6) and Eq. (7) to construct the ensembled affinity matrix $\hat{\mathbf{A}}$ except that $\boldsymbol{\theta}\_0=vec\left(\bar{\mathbf{W}}\right)=vec\left(\frac{1}{B}\sum\_{b=1}^B\mathbf{W}^{(b)}\right)$ where, following Prompt Ensembling used in TAC, $\mathbf{W}^{(b)}$ is the CLIP features of positive nouns induced by $b$-th prompt template $\Delta^{(b)}(\cdot)$. Formally, $\hat{\mathbf{A}}$ computed by the second alternative is given by:
>
> $$
> \hat{\mathbf{A}}[i,j]= \\begin{cases}
> \mathcal{K}\_{\bar{\mathbf{W}}}\left(f\_{\mathcal{X}}(\mathbf{x}\_i),f\_{\mathcal{X}}(\mathbf{x}\_j)\right),  & \text{ if } f\_{\mathcal{X}}(\mathbf{x}\_i)\in\mathcal{N}\_{q}(f\_{\mathcal{X}}(\mathbf{x}\_j),\mathcal{K}\_{\bar{\mathbf{W}}})\wedge f\_{\mathcal{X}}(\mathbf{x}\_j)\in\mathcal{N}\_{q}(f\_{\mathcal{X}}(\mathbf{x}\_i),\mathcal{K}\_{\bar{\mathbf{W}}}). \\\\
> 0, &\text{  otherwise. }
> \\end{cases}
> $$
>
> The tables below show that Ours (full model) achieves the best clustering performance, which validates the effectiveness of RAD. Besides, one can also find that Ours (b) significantly outperforms TAC, which implies the effectiveness of our proposed NTK-induced affinity measure.
>
> | CIFAR-10| NMI | ACC | ARI |
> |-|-|-|-|
> | TAC |    80.8      |    90.1        |   79.8
> |  Ours (a)  |71.4|74.7|56.8
> | Ours (b) |82.9| 91.3| 82.4
> |  Ours (full model)       | 83.3           |     92.0       |   83.0
>
> | CIFAR-20| NMI | ACC | ARI |
> |-|-|-|-|
> | TAC |60.7|55.8|42.7
> |  Ours (a)  |45.1|42.6|29.9
> | Ours (b) |60.9|55.0|43.4
> |  Ours (full model)       |63.3|59.6|43.5
>
> | ImageNet-1K| NMI | ACC | ARI |
> |-|-|-|-|
> | TAC |77.8|48.9|36.4
> |  Ours (a)  |77.1|51.0|35.4
> | Ours (b) |79.2|53.3|38.4
> |  Ours (full model) |79.2|56.3|39.4
>
> ## 3.1. Prompt templates
>
> To investigate how much has been the ensemble contributing to the performance compared to using only one prompt, we report the clustering performance under single-prompt setting as follows. Note that, when single prompt is used, our proposed RAD is automatically disabled.
>
> ## 1. **itap of a $\<label\>$**
>
> | CIFAR-10| NMI | ACC | ARI |
> |-|-|-|-|
> | TAC |76.9|80.7|70.5
> |  Ours |83.2|90.6|81.1
> | TAC (full model)|    80.8      |    90.1        |   79.8
> |  Ours (full model)  | 83.3           |     92.0       |   83.0
>
> | CIFAR-20| NMI | ACC | ARI |
> |-|-|-|-|
> | TAC |56.8|48.7|36.3
> |  Ours   |62.4|56.3|39.9
> | TAC (full model) |60.7|55.8|42.7
> | Ours (full model)   |63.3|59.6|43.5
>
> | ImageNet-1K| NMI | ACC | ARI |
> |-|-|-|-|
> | TAC |   75.1|46.4|33.6
> |  Ours |78.6|54.6|38.9
> | TAC (full model) |77.8|48.9|36.4|
> |  Ours (full model)  |79.2|56.3|39.4
>
> ## 2. **a bad photo of the $\<label\>$**
>
> | CIFAR-10| NMI | ACC | ARI |
> |-|-|-|-|
> |TAC |81.8|90.0|70.5
> |Ours |83.1|91.6|82.1
> |TAC (full model)|80.8|90.1|79.8
> |Ours (full model)| 83.3|92.0|83.0
>
> |CIFAR-20| NMI | ACC | ARI |
> |-|-|-|-|
> |TAC|60.8|54.6|41.1
> |Ours|62.3|56.4|40.9
> |TAC (full model) |60.7|55.8|42.7
> | Ours (full model)|63.3|59.6|43.5
>
> |ImageNet-1K| NMI | ACC | ARI |
> |-|-|-|-|
> |TAC|75.9|47.6|36.0
> |Ours|78.9|55.6|39.4
> |TAC (full model)|77.8|48.9|36.4|
> |Ours (full model)|79.2|56.3|39.4
>
> ## **3. a origami $\<label\>$**
>
> | CIFAR-10| NMI | ACC | ARI |
> |-|-|-|-|
> |TAC| 79.6|87.9|74.8
> | Ours |82.6|91.1|81.7
> |TAC (full model)| 80.8|90.1|79.8
> | Ours (full model) |83.3  |92.0| 83.0
>
> | CIFAR-20| NMI | ACC | ARI |
> |-|-|-|--|
> |TAC| 58.4|51.3|37.2
> |Ours|63.6|58.7|40.3
> |TAC (full model) |60.7|55.8|42.7
> |Ours (full model)|63.3|59.6|43.5
>
> | ImageNet-1K| NMI | ACC | ARI |
> |-|-|-|-|
> |TAC|72.2|38.4|16.8
> |Ours|76.5|53.7|33.1
> |TAC (full model) |77.8|48.9|36.4|
> |Ours (full model)|79.2|56.3|39.4
>
> Several observations can be found in the tables above:
> - Even with a single prompt template, ours is still outperforms TAC (full model) and achieves comparable performance with Ours (full model), which implies that the performance gains do **NOT** donimately come from prompt engineering.
> - Compared with TAC, the clustering performance of ours is **less sensitive** to the choice of a single prompt template.
> - Ours consistently and significantly outperforms TAC (where the kernel way is not used), which implies the advantage of our proposed NTK-based graph construction still holds with a single prompt template.
>
> More experiment results regarding templates and datasets can be found in Appendix J of the revised manuscript.
>
> # 4. Some typos can be fixed, e.g., "aim to grouping".
>
> Thanks for pointing out the typo. We have revised the manuscript to fix all typos.

---

> > ### Author Response · Authors · 2025-11-25
> >
> > Dear Reviewer,
> >
> > The rebuttal phase has opened, and we have submitted a detailed response to your comments. Thank you again for your valuable feedback and for the time you have devoted to our paper.
> >
> > When convenient, **could you please take a look at our rebuttal and let us know if our responses adequately address your concerns? If you feel that the issues have been resolved, we would be sincerely grateful if you could consider increasing your score accordingly.**
> >
> > Your feedback would be extremely helpful for us to further refine the work.
> >
> > Thank you again for your time and thoughtful review.
> >
> > Kind regards,
> >
> > Senior Author

---

> ### Author Response · Authors · 2025-11-26
>
> Dear Reviewer,
>
> The rebuttal phase has opened, and we have submitted a detailed response to your comments. Thank you again for your valuable feedback and for the time you have devoted to our paper.
>
> When convenient, could you please take a look at our rebuttal and let us know if our responses adequately address your concerns? If you feel that the issues have been resolved, we would be sincerely grateful if you could consider increasing your score accordingly.
>
> Your feedback would be extremely helpful for us to further refine the work.
>
> Thank you again for your time and thoughtful review.
>
> Kind regards,
>
> Senior Author

---

### Author Response · Authors · 2025-11-21

Dear Chairs and Reviewers,

We sincerely thank you for your time and efforts throughout the reviewing period. We have uploaded the revised version and responded to all the reviewers in detail. We believe that the quality of the paper is improved and the contributions are solid. In particular, we would like to highlight some key materials we added:

1. Ablation Study on Prompt Templates (Appendix J)
2. Validation of Image and Class Name Leakage (Appendix K and Appendix L)
3. Hypothesis Testing for Evaluating Statistical Significance (Appendix M)
4. Analsysis on Time and Space Complexity of Our Method (Appendix N).

We understand that reviewers are busy during the response period, we would greatly appreciate it if the reviewers could kindly advise if our responses solve their concerns. If there are any other suggestions/questions, we will try our best to provide satisfactory answers. We are looking forward to any further discussion with the reviewers.

Best regards,

The authors

---

### Meta-Review · Area_Chair_apUw · 2025-12-15

**Summary:**

The paper initially received mixed reviews (scores 4, 4, 6, 8). The primary concerns focused on potential data leakage due to the overlap between CLIP's training data and the evaluation benchmarks, the handling of negative values in the affinity matrix, and questions regarding scalability and computational complexity. The authors provided a robust rebuttal, including new experiments with OpenCLIP to rule out data leakage, scalability analysis on ImageNet-1K, and clarifications on normalization. Given the strong empirical evidence and the resolution of technical queries, the recommendation is to accept.

**Reviewer Concerns:**

- Data Leakage (Reviewer KbNX): Effectively addressed by additional experiments using OpenCLIP (which has minimal data overlap), demonstrating that performance gains are not due to leakage.
- Scalability & Complexity (Reviewers nmSD, kiq4): Addressed by providing time/space complexity analysis and reporting actual runtimes on ImageNet-1K.
- Negative Affinities (Reviewer nmSD): Addressed by clarifying the use of min-max normalization to ensure non-negative inputs for spectral clustering.
- Ablations & Baselines (Reviewers KbNX, kiq4): Addressed by adding ablation studies on the RAD module and comparing with training-based TAC variants.
- Reviewer kiq4 explicitly confirmed that all their concerns were addressed.

**Reviewer Scores:**

- Reviewer KbNX (Original: 4 -> Estimated: 6): The reviewer's main concern about data leakage was empirically disproven.
- Reviewer nmSD (Original: 4 -> Estimated: 6): The technical concerns regarding negative weights and graph robustness were clarified with specific implementation details and sensitivity analyses.
- Reviewer rFTU (Original: 8 -> Estimated: 8): Remained positive; the rebuttal addressed the minor bias concern.
- Reviewer kiq4 (Original: 6 -> Estimated: 7): Explicitly stated concerns were addressed, justifying a score increase.

---

### Decision · Program_Chairs · 2026-01-26

Accept (Poster)